# Machine learning estimates of eddy covariance carbon flux in a scrub in the Mexican highland

Aurelio Guevara-Escobar[1], Enrique González-Sosa[2], Mónica Cervantes-Jiménez[1], Humberto Suzán-Azpiri[1], Mónica Elisa Queijeiro-Bolaños[1], Israel Carrillo-Ángeles[1], Víctor Hugo Cambrón-Sandoval[1].

[1]Facultad de Ciencias Naturales, Universidad Autónoma de Querétaro, Av. de las Ciencias s/n Juriquilla CP. 76230, Querétaro, Querétaro.
[2]Facultad de Ingeniería, Universidad Autónoma de Querétaro. Cerro de las Campanas s/n Las Campanas, CP. 76010 Querétaro, Querétaro.

*Correspondence to*: Mónica Cervantes-Jiménez (monica.cervantes@uaq.mx)

**Abstract.** Arid and semi-arid ecosystems contain relatively high species diversity and are subject to intense use, in particular extensive cattle grazing which has favoured the expansion and encroachment of perennial thorny shrubs into the grasslands, thus decreasing the value of the rangeland. However, these environments have been shown to positively impact global carbon dynamics. Machine learning and remote sensing had enhanced our knowledge about carbon dynamics, but they need to be further developed and adapted to particular analysis. We measured the net ecosystem exchange of C (NEE) with the Eddy Covariance (EC) method and estimated GPP in a thorny scrub at Bernal in Mexico. We tested the agreement between EC estimates and remotely sensed GPP estimates from MODIS, and also with two alternative modeling methods: ordinary least squares multiple regression (OLS) or ensembles of machine learning algorithms (EML). The variables used as predictors were MODIS spectral bands, vegetation indices and products, as well as gridded environmental variables. The Bernal site was a carbon sink despite it being in an overgrazed condition, the average NEE during fifteen months of 2017 and 2018 was -0.78 g C $m^{-2}$ $d^{-1}$ and the flux was negative or neutral during the measured months. The probability of agreement ($\theta$s) represented the agreement between observed and estimated values of GPP across the range of measurement. According to the mean value of $\theta$s, agreement was higher for the EML (0.6) followed by OLS (0.5) and then MODIS (0.24). This graphic metric was more informative than $r^2$ (0.98, 0.67, 0.58 respectively) to evaluate the model performance. This was particularly true for MODIS because the maximum $\theta$s of 4.3 was for measurements of 0.8 g C $m^{-2}$ $d^{-1}$ and then decreased steadily below 1 $\theta$s for measurements above 6.5 g C $m^{-2}$ $d^{-1}$ for this scrub vegetation. In the case of EML and OLS the $\theta$s was estable across the range of measurement. We used an EML for the Ameriflux site US-SRM, which has similar vegetation and climate, to predict GPP at Bernal but $\theta$s was low (0.16), indicating the local specificity of this model. Although cacti were an important component of the vegetation, the nighttime flux was characterized by positive NEE suggesting that the photosynthetic dark-cycle flux of cacti was lower than ecosystem respiration. The discrepancy between MODIS and EC GPP estimates stresses the need to understand the limitations of both methods.

# 1 Introduction

Deserts and semi deserts occupy more than 30% of terrestrial ecosystems, but almost 2 million km$^2$ (50%) correspond to arid and semi-arid ecosystems in Mexico, mainly the Sonoran and the Chihuahuan deserts (Verbist et al., 2010). The Spanish-Criollo intrusion (1540-1640) brought new land-use methods, but there is no evidence of additional landscape degradation from the central highlands to the north-eastern frontier of New Spain until well into the 18[th] century (Butzer and Butzer, 1997). At the country scale, the extent of grasslands in Mexico declined and the area of croplands and woody areas increased; rural-urban migration being an important driver of that transition (Bonilla-Moheno and Aide, 2020). The transition from grasslands to shrublands or scrub is linked to the extremely heavy grazing by domestic livestock (Wilcox et al., 2018).

Vegetation in the arid and semi-arid ecosystems are mostly classified as rangelands. These are one of the most widely distributed landscapes on earth, incorporating a wide range of communities including grasslands, shrublands, and savannah. Scrub is a xeric category of shrublands characterized by plants with small leaves, very thorny and its biomass is distributed mainly to roots and leaves rather than the stems (Rzedowski, 1978; Wheeler et al., 2007; Zhang et al., 2017). Studies based on multiple sources of evidence, considering the driving processes of the land-use change in Mexico are needed to aid in policy formulation and to identify regions that may provide important ecosystem services (Murray-Tortarolo et al., 2016).

On the other hand, photosynthesis contributes to carbon sequestration by moving carbon stock from the atmosphere to other pools or sinks, as above ground biomass, roots and soil organic matter (Booker et al., 2013). The role of vegetation in carbon sequestration on arid and semi-arid ecosystems is less evident because the growth rate is low and biomass partition above and below ground is different from that of temperate and tropical forests. The competitive interactions of arid plants at the community level are strongly influenced by rooting architecture and phenological growth (Zeng et al., 2008). Many plants in semi-arid systems support a deep and wide root system as a drought adaptation but also for nutrient uptake (McCulley et al., 2004).

Recent time-trends indicate that semi-arid ecosystems regulate the terrestrial carbon sink and dominate its inter-annual variability (Piao et al., 2019; Scott et al., 2015; Zhang et al., 2020). This variability mainly results from the imbalance between two larger biogenic fluxes that constitute the net ecosystem exchange (NEE): the photosynthetic uptake of $CO_2$ (gross primary production, GPP) and the respiratory release of $CO_2$ (total ecosystem respiration, $R_{eco}$). Radiation and water availability are important environmental drivers of NEE and thus, GPP and $R_{eco}$ (Marcolla et al., 2017). However, other carbon fluxes contribute to the imbalance, such as fire and anthropogenic $CO_2$ emissions (Järvi et al., 2019; Piao et al., 2019). Another atmospheric $CO_2$ flux is that from soil inorganic carbon in arid and semi-arid ecosystems (Soper et al., 2017). Calcium carbonates form in the soil at a relatively low rate of 5 to 150 kg C ha$^{-1}$ y$^{-1}$; this carbon can return to the atmosphere, but they are a carbon sink when carbonates are leached into the groundwater (Lal et al., 2004).

The methods used to explore the ecosystems and the understanding of their functioning is changing rapidly, particularly for arid and semi-arid ecosystems (Goldstein et al., 2020; Ma et al., 2020; Xiao et al., 2019; Yao et al., 2020). There are many instruments and techniques for estimating carbon and water fluxes but two stands out in the literature: Eddy Covariance (EC)

and remote sensing techniques. EC is a micro-meteorological method that measures the ecosystem community NEE at short time intervals representing a land surface smaller than 1 km$^2$. The orbital remote sensors measures radiation emitted or reflected by earth surfaces and; using different algorithms, represents different traits of vegetation activity from large-scale areas. Both techniques are complementary, but an agreement between their estimates is important for regional, countrywide or global spatiotemporal monitoring of greenhouse gas inventories (Yona et al., 2020), ecological modeling, quantifying the interaction among the vegetation component and the hydrological, energy and nutrient cycles; among others applications (Pasetto et al., 2018). Particularly, products from the moderate resolution spectroradiometer (MODIS) have ample availability and are extensively used to study land surface since 2000.

Gross primary production can be represented by a wide range of models, ranging in complexity from simple regression based on climatic forcing variables to complex models that simulate biophysical and ecophysiological processes (Anav et al., 2015). The MODIS MOD17 product uses a photosynthetic radiation conversion efficiency model (Running and Zhao, 2015), but a better relationship is reported with EC derived GPP when the model uses vegetation indices calculated from the same MODIS platform (Ma et al., 2014; Wu et al., 2010). Although they are black box models in principle, recent modeling efforts report good agreement of GPP estimates obtained from machine learning algorithms (ML) or ensembles of models (Eshel et al., 2019; Joiner and Yoshida, 2020; Jung et al., 2019). Different machine learning algorithms are powerful because they can identify trends and patterns in big data sets and solve regression or classification problems.

To generate models of GPP we measured EC fluxes during 2017-2018 in a thorny scrub with semi-arid climate in the highlands of Mexico (Bernal site). Competing models were data-driven machine learning regression ensembles (EML) and ordinary least squares regression (OLS), both using Daymet (Thornton et al., 2017) and MODIS data sets as explanatory variables. The MODIS GPP product was used as a baseline comparison. The second step was to use an EML model based on local data (Daymet and MODIS) from a site with EC instrumentation and similar vegetation to that of Bernal's site and then use that model to predict GPP at the Bernal site. The site we used was Santa Rita from the Ameriflux network. While Santa Rita is in the Sonoran Desert and Bernal is in the southern border of the Chihuahuan desert, both have a similar climate and vegetation (Figure 1). A good agreement between Bernal EC data and the predictions from Santa Rita model, would support the use of machine learning algorithms as a scale up mechanism. This would be useful to the understanding of semi-arid ecosystems and also improve current earth system models (Piao et al., 2019).

We measured the carbon flux for the vegetation in a semiarid site located at Bernal, Mexico. The phenology patterns in the region suggested that this site could be a carbon sink during the wet season or a carbon source during the dry season, since some of predominant species reproduce during winter and spring, particularly cacti, *Acacia* and *Prosopis* (Mesquite). Furthermore, the Bernal site had a history of disturbance by overgrazing, this could decrease the GPP and even result in a positive carbon balance; thus being a carbon source. If the shrub vegetation in this site predominantly absorbed carbon during the measuring period then this evidence would contribute to reinforce the reported importance of semi-arid environments in the global carbon balance (Zhang et al., 2020). However, land ownership patterns and the balance between agricultural investments and land conservation will determine the absolute amount of carbon sequestered. Hopefully, the results of the

present investigation would promote the idea that carbon sequestration is possible in scrubland and this be incorporated in informed decisions and new policy.

## 2 Materials & Methods

### 2.1 Site description

The study site (Bernal) is located at N 20.717, W 99.941 and 2,050 m a.s.l. in the municipality of Ezequiel Montes in Querétaro where real estate development, feedlot beef production, cheese and wine production associated with tourism and, automotive industry development are very attractive options for landowners in the region. Bernal is located in a shallow valley oriented from north to south, approximately 15 to 20 km wide and opening to the south to the Río Lerma basin and then draining into the Pacific Ocean. The northern limit of the valley is surrounded by hill country and its characteristic 433 m in height dacitic dome (Aguirre-Díaz et al., 2013). Moisture-laden winds blow westward from the Gulf of Mexico but the Sierra Gorda, located 60 km east of Bernal, casts a rain shadow over the area (Segerstrom, 1961).

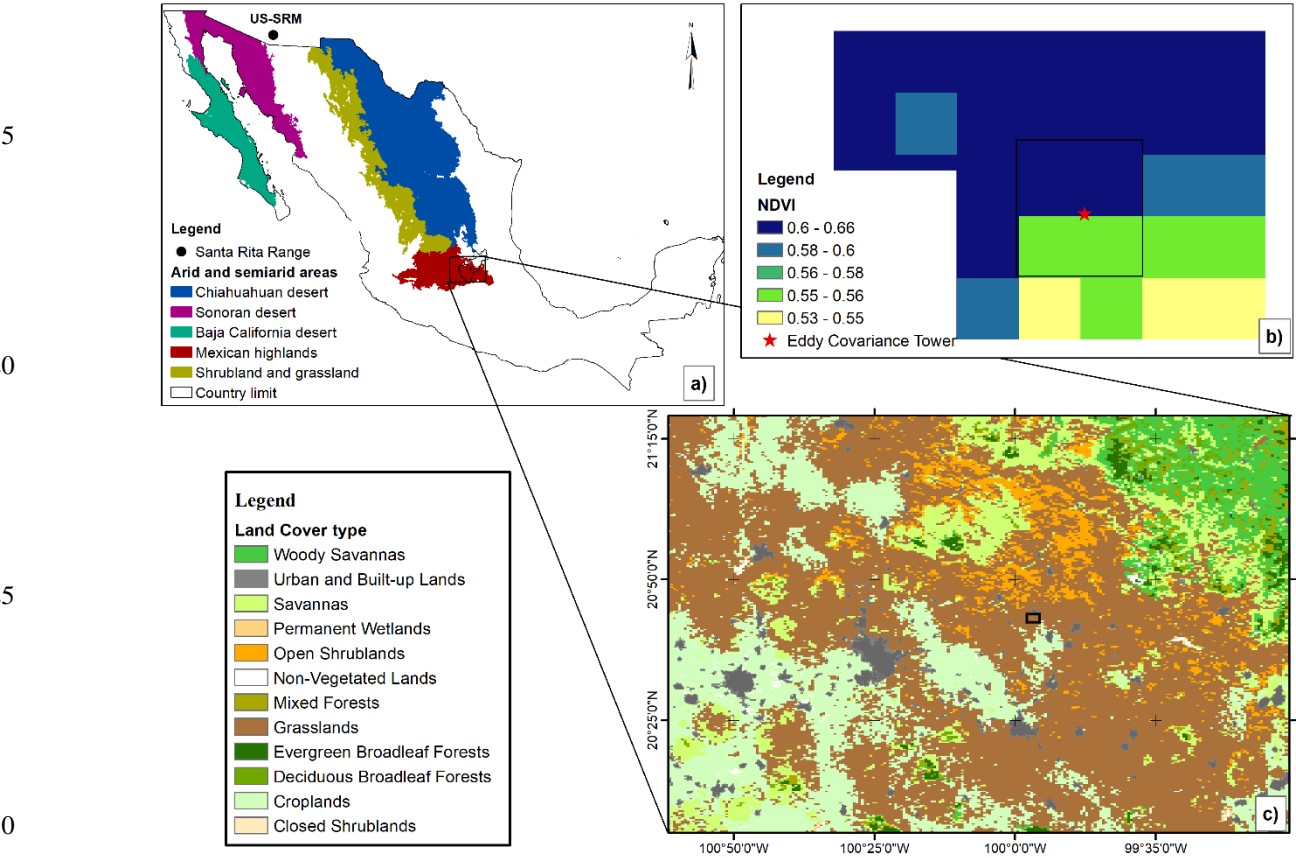

**Figure 1. Localization and land use maps for the study site: a) Biogeographic arid and semi-arid zones of southern North America relevant for the Bernal and Santa Rita sites, the black outline is the state limit for Queretaro b) Heterogeneity of the normalized difference vegetation index (NDVI) surrounding the EC tower at Bernal during the peak of the growing season 2017 (DOY 257), c) Land cover in the region of Bernal site, according to the Annual University of Maryland (UMD) classification (MCD12 MODIS product).**

The Bernal site is a private property, with grazing dairy cattle receiving additional concentrated foodstuffs under stall-feeding. Grazing was continuous, water for livestock was only available in the feeding and milking area; there were no pasture divisions and the perimeter fence was made of stone. These characteristics of the animal production model and the state of vegetation are representative of land management practices and the scrub vegetation in this region. However, the Bernal site suffered important changes in land use during 2019 and the scrub was suddenly cleared and converted into rainfed cropping.

The climate is arid with summer rains (BSk), mean annual rainfall is 476 mm and mean annual temperature is 17.1 °C (CICESE, 2015). Prevailing wind is from the east and north-east. The terrain is mostly flat; most grades are below 2%. The soil has a clay loam texture, the class is a Vertisol with abundant sub rounded basaltic stones, without rocky outcrops and the depth is greater than 0.6 m. The vegetation is less than 3 m in height with an overgrazed herbaceous stratum. The vegetation corresponds to secondary scrub with the dominant genera *Acacia*, *Prosopis* and different Cacti (Figure 3). This site was classified as grassland by MODIS landcover product.

For the scrub and tree species, the importance vegetation index (IVI) was determined following Curtis and McIntosh (1950) to assess the vegetation homogeneity. The IVI is the sum of relative dominance, relative density and relative frequency of the species present. Vegetation sample points were chosen according to the flux footprint of the eddy covariance tower (Figure 2). For each plant in the vegetation sample points, two stem diameters, the number of individuals (abundance) and identity of each species were measured, as well as the coverage, which is the horizontal projection of the aerial parts of the individuals on the ground, expressed as a percentage of the total area (Wilson, 2011).

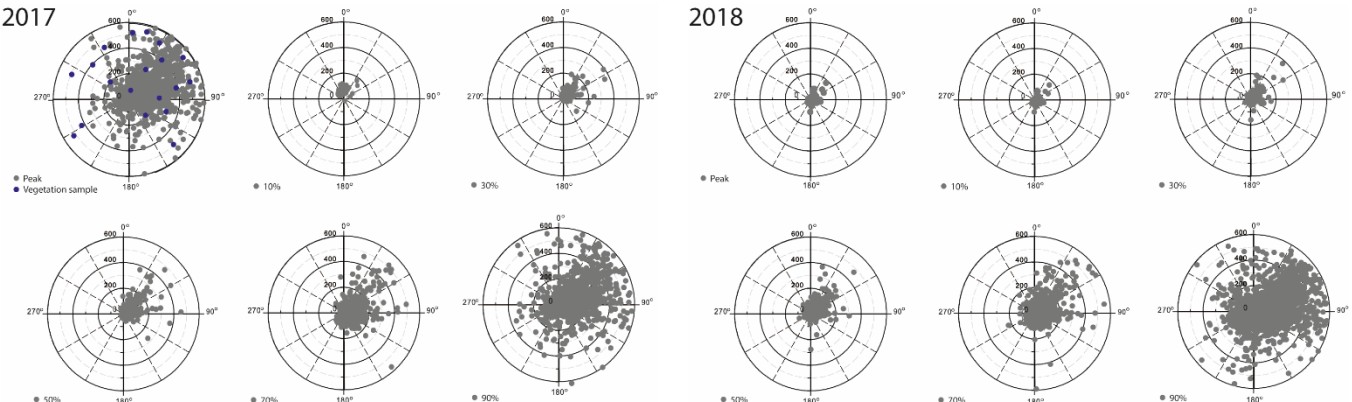

**Figure 2. Eddy covariance flux footprint at Bernal site during 2017 and 2018. Percentages are the contribution fluxes according to wind direction. Distance scale are meters (0-600). Blue points represent the vegetation sample plots used to calculate the Importance Vegetation Index (IVI).**

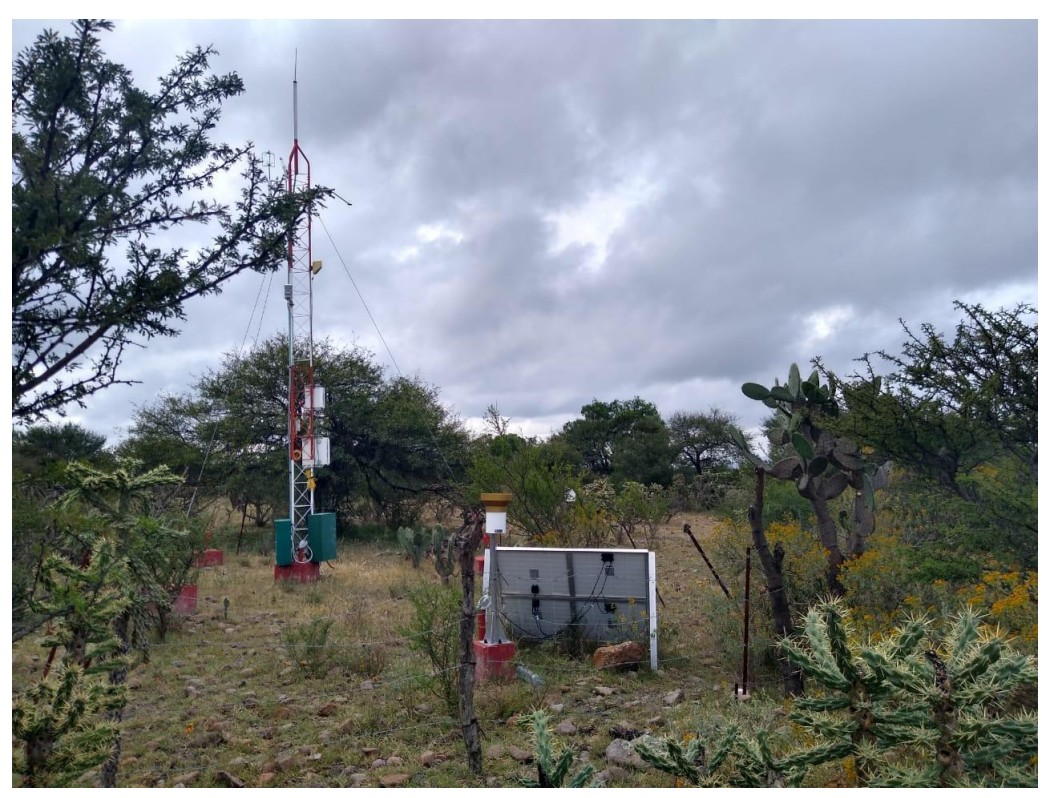

**Figure 3. Thorny scrub at Bernal, Queretaro during the rainy season 2017. In the foreground *Cylindropuntia imbricata*, a very thorny cactus, shrubs in the background are *Prosopis laevigata* mesquites.**

### 2.2 Eddy covariance measurements

The micro-meteorological EC technique, measures at the plant community level NEE in a non-destructively way, and

10    continuously over time (Baldocchi, 2014). The negative $CO_2$ fluxes corresponded to NEE, which is equivalent to NEP (net ecosystem production) but with opposite sign. The EC has advantages compared to other techniques that need to scale up measurements from the leaf, plant or soil levels up to ecosystems, especially when the vegetation is heterogeneous (Yepez et al., 2003). However, EC is an expensive technique, data analysis and processing are complicated, also specific assumptions must be met regarding the terrain, vegetation and micro-meteorological conditions, among other aspects (Richardson et al.,

15    2019).

The fluxes were measured with the EC technique at a height of 6 m with the following instruments: A Biomet system (Licor Biosciences, USA) to measure $H_2O$ and $CO_2$ fluxes using an IRGASON-EC-150 open circuit analyzer, and a CSAT3 sonic anemometer, KH20 krypton hygrometer, these were connected to a CR3000 datalogger (Campbell Scientific Inc., Logan, UT, USA). The relative humidity and air temperature were measured with an HMP155A probe (Vaisala Corporation, Helsinki, Finland); net radiation was measured with a NR-Lite2 radiometer (Kipp & Zonen BV Delft, The Netherlands); and the photosynthetic active radiation (PAR) was measured with a quantum sensor SKP215 (Skye Instruments, Llandrindod Wells, UK). Measurements of the soils heat flux was implemented with four self-calibrating HFP01SC plates at 80 mm depth and in four representative positions of the landscape (Hukseflux Thermal Sensors BV, Delft, The Netherlands). Three time domain reflectometry probes (TDR) CS616 measured volumetric water content in the ground installed vertically, and two sets of TCAV thermocouples measured the temperature at 60 and 40 mm depths and above the HFP01SC plates (Campbell Scientific Inc., Logan, UT, USA). The TE525 (Texas Electronics, Dallas, TX, USA) tipping bucket rain gauge was installed at 1.2 m high and three meters away from the tower. All these meteorological variables were measured every 5 seconds and average values were stored every 30 minutes; rainfall was accumulated for the same time interval. Sensible (H) and latent (λE) heat fluxes calculated by the EddyPro package (Licor Biosciences, USA).

**2.3 Flux data processing**

All EC data were collected at 10 Mhz in the datalogger and reported as μmol $CO_2$ $m^{-2}$ $s^{-1}$ and processed with the Eddypro package to convert values into average fluxes of 30-minute intervals. Only quality flagged records were used to account for the $CO_2$ flux (qc_co2_flux = 0) according to the Mauder and Foken (2011) policy in the Eddypro program (Licor, 2019). However, this quality-checking is not sufficient especially in the case of $CO_2$; therefore, data was post processed using the Reddyproc package of R (R Development Core Team, 2009), to estimate the friction speed thresholds (u*), gap-fill data, and partition the NEE flux into its GPP and $R_{eco}$ components (Wutzler et al., 2018). The filled-in estimates of NEE (NEE_uStar_f), GPP (GPP_uStar_f), and $R_{eco}$ (Reco_uStar) were used when the u* was lower than an u* annual threshold above which night-time fluxes are considered valid. The annual u* threshold was 0.193 and 0.194 for 2017 and 2018. The difference between these thresholds and the 95% u* threshold was small (0.033 m $s^{-1}$). Appendix A presents the threshold means and confidence intervals calculated in the Reddyproc package. Only 1050 half hour records (9.3%) had an u* below the annual mean u* threshold. The data with a flag equal to 0 was used for the variable NEE_uStar_fqc, as defined by Reddyproc. Carbon dioxide flux data were time integrated and converted to g C $m^{-2}$ $d^{-1}$ using the molar ratio of C. We only reported the continuous measurements of the exchange of $CO_2$ for the period of April 2017 (DOY 89) to August 2018 (DOY 234) using the EC technique. Due to equipment malfunction and incomplete datasets some periods of time were not considered. The measurement campaign presented here was not biased by wet winters, since both years were characterized by a less than weak Niño-Niña.

## 2.4 Remote sensed data

Data was requested via the land processes DAAC AppEEARS to obtain spatial and temporal subsets for the Bernal and Santa Rita sites including: daily surface reflectance (MOD09GA.006 and MODOCGA.006), daily day and night time land surface temperature (LST) (MOD11A2.006 and MYD11A2.006), eight day leaf area index (LAI) and fraction of photosynthetically active radiation (FPAR) (MOD15A2H.006, MYD15A2H.006, MCD15A2H.006), sixteen day enhanced vegetation index (EVI) (MYD13A1.006), sixteen day gross primary production (GPP) and net photosynthesis (PsnNet) (MOD17A2H.006). The AppEEARS also unpacks and interprets the quality layers. Appendix B presents the details of each spectral band of MODIS. Data with less than good quality flags were deleted. Missing data was filled with splines and a database with one day time step was generated, this would smooth linear temporal phenological evolution between any two successive remotely sensed data points (Eshel et al., 2019). Daily accumulated rainfall was requested using the Giovanni GSFC platform (GPM 3IMERGDF.006 and TRMM 3B42.007). Gridded weather parameters from de ORNL DAAC Daymet dataset were: precipitation, shortwave radiation, maximum and minimum air temperature and water vapor pressure. Daymet is a data product derived from a collection of algorithms interpolating and extrapolating daily meteorological observations (Thornton et al., 2017). Following Henrich et al. (2012) and Hill et al. (2006), daily reflectance bands of MODIS were used to compute several vegetation indices: Red Green Ratio Index (RGRI), Simple Ratio (SimpleR), Moisture Stress (MoistS), Disease Stress Index (DSI), Normalized Difference Vegetation Index (NDVI), Normalized Difference Water Index (NDVI_w) and Enhanced Vegetation Index (EVI); the corresponding equations are presented in Appendix B.

## 2.5 MODIS algorithm for GPP

Estimates of GPP are derived from data recorded by the MODIS sensor aboard the Terra and Aqua satellites. The efficiency ($\varepsilon$, g C MJ$^{-1}$) with which vegetation produces dry matter is defined as the amount of solar energy stored by photosynthesis in a given period, divided by the solar constant integrated over the same period (Monteith, 1972). Not all incident solar radiation is available for biomass conversion, only about 48% is photosynthetically active (PAR, MJ m$^{-2}$) and not all PAR is absorbed (Zhu et al., 2008). Thus, carbon exchange is mainly controlled by the amount of PAR absorbed by green vegetation (APAR) and modified by $\varepsilon$ (Gitelson et al., 2015). The fraction of absorbed PAR (FPAR) is equal to APAR/PAR, but can be represented by the NDVI spectral vegetation index produced by MODIS (Running 2004). The efficiency term $\varepsilon$ is described as the product of different factors as a whole or part of the system (Monteith, 1972), but mostly those related to the efficiencies with which the vegetation intercepts the radiation and the efficiency to convert the intercepted radiation into biomass (Long et al., 2015). The MODIS algorithm that estimates GPP in the MOD17 product is (Running et al., 2004):

$$GPP = \varepsilon \times FPAR \times PAR \qquad\qquad \text{Eq. (1)}$$
$$GPP = \varepsilon \times NDVI \times PAR \qquad\qquad \text{Eq. (2)}$$

The ε term in the MODIS algorithm is represented by a maximum radiation conversion efficiency ($\varepsilon_{max}$, kg C MJ$^{-1}$) that is attenuated by sub optimal climatic conditions, mainly minimum air temperature (Tmin) and vapor pressure deficit (VPD). Two parameters for each, Tmin and VPD, are used to define attenuation scalars for general biome types. These parameters form linear functions between the scalars (Running and Zhao, 2015; Wang et al., 2013): daily minimum temperature at which ε = $\varepsilon_{max}$ and at which ε=0 and; the daylight average VPD at which ε = $\varepsilon_{max}$ and at which ε=0. GPP is truncated on days when air temperature is below 0°C or VPD is higher than 2000 Pa (Running and Zhao, 2015). Stress and nutrient constraints on vegetation growth are quantified by the limiting relation of leaf area in NDVI x PAR, rather than constrained through ε (Running et al., 2004). However, the MODIS algorithm does not consider stomatal sensitivity to leaf-to-air vapor pressure across and within species and particularly between isohydric and anisohydric plant species (Grossiord et al., 2020).

$$\varepsilon = \varepsilon_{max} \times Tmin_{scalar} \times VPD\_scalar \qquad \text{Eq. (3)}$$

The MOD17 user's guide presents a Biome-Property-Look-Up-Table (BPLUT) with the parameters for each biome type and assumes that they do not vary with space or time (Running and Zhao, 2015). This aspect is important because $\varepsilon_{max}$ has the strongest impact on the predicted GPP of the MOD17 algorithm (Wang et al., 2013). The assumption also is important because the overstory and understory could be decoupled from each other and would intercept different amounts of light and have different water sources during the growing season (Scott et al., 2003). Light quantity and quality as diffuse light or sunflecks determine differences among understorey species in their temporal response to gaps, involving acclimation and avoidance of photoinhibition (Pearcy, 2007). Another shortcoming is that few land cover classifications are incorporated into the MYD17 algorithm.

## 2.5 Santa Rita site dataset

Santa Rita Experimental Range (SRER) is located in the western range of the Santa Rita Mountains in Arizona, USA (31.8214 latitude, -110.8661 longitude, 1,120 m asl.). Climate is BSk with mean annual precipitation of 380 mm, temperature of 17.9°C and Ustic Torri fluvents soils. Established in 1903, SRER has a long history of experimental manipulations to enhance grazing potential for cattle (Glenn et al., 2015). Two Ameriflux sites are located in the SRER: Santa Rita Grassland (US-SRG) and Santa Rita Mezquite (US-SRM). We used EC data for the years 2013-2019 from US-SRM which is a mesquite grass savanna (35% mesquite canopy cover and mean canopy height above 2m, 22% grasses and 43% bare soil), although MODIS describes this site as open shrublands (Glenn et al., 2015; Scott et al., 2004). The US-SRM site is dominated by velvet mesquite (*Prosopis velutina*), has a diversity of shrubs, cacti, succulents and bunch grasses (McClaran, 2003). This site was chosen because the vegetation and climate are similar to the Bernal site and it was the closest EC instrumentation with data availability (Figure 4).

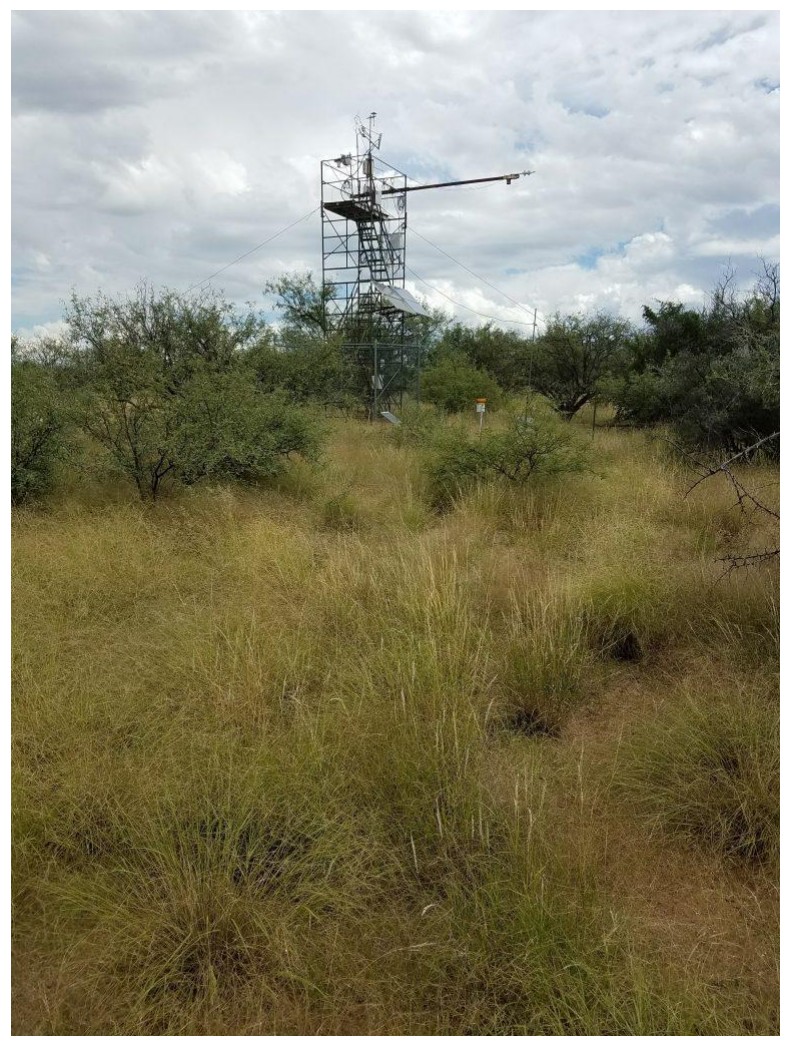

**Figure 4. Semidesert grassland encroached by mesquite (*Prosopis velutina*) at Santa Rita, Arizona (US-SRM). Image Credit: Russell Scott, 09/12/2016.**

### 2.6 Modeling

Gross primary production estimated by EC at Bernal site was modeled using OLS and EML. The explanatory variables were the remote sensed data, the weather parameters and the vegetation indices (Appendix B). The OLS is a particular case of the generalized linear model where the variation of single response variable is explained by several independent variables. The OLS was fitted with the stepwise procedure, the final model included variables with a variance inflation factor (VIF) lower than 10 and a significance level of 0.05. Predictions of the EC GPP were obtained with the final model. Analysis and diagnostics were made with Minitab v 17 (Minitab LLC). Analysis of the OLS model can be used to determine agreement between methods of measurement, but it is sensitive to the range of values in the dataset and its metrics, r, $r^2$ and root mean

square, do not provide information on the type of association (Bland and Altman, 2010). In this paper, we used another metric, the probability of agreement, to determine bias and agreement between model estimates and observed data (see below).

While OLS is a well-known algorithm, machine learning algorithms are emerging techniques that focus on the data structure and match that data onto models. The EML approach considers the different realizations of machine learning models and constructs an ensemble models coming with the advantage of being more accurate that the predictions from the individual ensemble members. However, EML is compute intensive requiring nodes with purpose-built hardware such as multiple processors or reduced-precision accelerators. The nodes could be aggregated in computing clusters which require storage, power and cooling redundancy.

A stack of EML was obtained with the H2O package of R (H2O.ai 2017). This package provides several algorithms that can contribute to a stack of ensembles using the automl function: feedforward artificial neural network (DL), general linear models (GLM), gradient boosting machine (GMB), extreme gradient boosting (XGBoost), default distributed random forest (DRF), extremely randomized trees (XRT) and general linear models (GLM). Automl trains two stacked ensemble models, one ensemble contains all the models, and the second ensemble contains just the best performing model from each algorithm class/family; both of the ensembles should produce better models than any individual model from the automl run. The term automl (Automatic Machine Learning) implies data preprocessing, normalization, feature engineering, model selection, hyperparameter optimization, and prediction analysis; including procedures to identify and deal with non-independent and identically distributed observations and overfitting (Michailidis, 2018; Truong et al., 2019).

Machine learning has two elements for supervised learning: training loss and regularization. The task of training tries to find the best parameters for the model while minimizing the training loss function; the mean squared error for example. The regularization term controls the complexity of the model helping to reduce overfitting. Overfitting becomes apparent when the model performs accurately during the training but the accuracy is low during the testing. A good model needs extensive parameter tuning by running many times the algorithm to explore the effect on regularization and cross validation accuracy (Mitchell and Frank, 2017). In this investigation the function of training loss was the deviance, which is a generalization of the residual sum of squares driven by the likelihood. Deviance is a measure of model fit, lower or negative values indicate better model performance (McElreath, 2020).

A stack of EML solutions was based on a random sample of the dataset for training the model. For the Bernal site 85% was used for training and 80% in the case of US-SRM. The automl function was run 20 times, each run added approximately 48 models to the leaderboard and ranked the best performing models by their deviance. Each run splits the training data ten times for k-fold cross-validation. The seed, for EML that is dependent on randomization, was changed in every run. The stopping rule for each run was set at 100 s and the maximum memory allocation pool for H2O was 100 Gb, in a single workstation with dual Xeon 2680 v4 processors and 128 Gb of RAM. The H2O package was installed in a rocker/geospatial docker container, which is a portable, scalable and reproducible environment (Boettiger and Eddelbuettel, 2017).

Two sets of predictions for the GPP at the Bernal site were obtained from the stacked ensemble. The first set of predictions was based on the 15% of the Bernal site data reserved for testing. The second set of predictions was obtained by re-feeding the

US-SRM site model with the Bernal site explanatory variables. The first set of predictions would show the importance of local data to predict EC based GPP. The second set of predictions would represent the suitability of off-site data to predict EC based GPP. If the second scenario has good agreement, then an EML model could be used to represent wider areas of the ecosystem.

## 2.7 Variable importance

The variable importance within individual models was used to answer the question of which environmental variables were important for GPP prediction. For the "all models ensemble" and "best of family ensemble" generated by automl is not possible to examine the variable importance nor the contribution of the individual models to the stack (H2O.ai 2017). Therefore, a weight ($w_i$) was calculated using equation (4), which is adequate for other information criterion besides the Akaike weights; this weight is an estimate of the conditional probability that the model will make best predictions on new data considering the set of models (McElreath, 2020). Then, the importance of each variable (%) was multiplied by the model's weight ($w_i$) and then added by variable to build the variable importance index. This index would measure how often a given variable was used in the leaderboard.

$$w_i = \frac{\exp\left(-\frac{1}{2} dWAIC_i\right)}{\sum_{j=1}^{m} \exp\left(-\frac{1}{2} dWAIC_i\right)}$$ Eq. (4)

$$dWAIC = deviance_i - deviance\ of\ top\ performing\ model$$ Eq. (5)

## 2.8 Model agreement

Calibration and agreement between methods of measurement are different procedures. Calibration compares known quantities of the true value or measurements made by highly accurate method (a gold standard), against the measurements of a new or a contending method. When two methods of measurement are compared neither provides an unequivocally correct measurement, because both have a measurement error and the true value remains unknown (Bland and Altman, 2010). Stevens et al. (2015) proposes the probability of agreement ($\theta s$) as a plot-metric to represent the agreement between two measurement systems across a range of plausible values. The $\theta s$ method addresses some of the challenges of the accepted "limits agreement method" presented by Bland and Altman (2010). Besides the agreement plot, agreement is based on maximum likelihood bias parameters: $\alpha$ and $\beta$ quantifying the fixed bias and the proportional bias. If $\alpha=0$, $\beta=1$ and $\sigma 1 = \sigma 2$ then the two measurements are identical; where $\sigma j$ are the measurement variation. The probability of agreement analysis was performed using the ProbAgreeAnalysis ([https://uwaterloo.ca/business-and-industrial-statistics-research-group/software](https://uwaterloo.ca/business-and-industrial-statistics-research-group/software)) in Matlab 9.4 (MathWorks, Inc.). An arbitrary 1 g C m$^{-2}$ d$^{-1}$ was considered as a tolerable magnitude to conclude that agreement is sufficient as to use either estimate GPP interchangeably. The reference measurement was the GPP obtained from EC data at Bernal site and tested against the MODIS MOD17 model, the OLS predictions or each of the two sets of EML predictions. If the

probability of agreement plot suggested disagreement between two measurement systems, then the predictions can be adjusted using:

$$Adjusted\ predictor = (predictor - \alpha)/\beta$$    Eq. (6)

## 3 Results

### 3.1 Eddy covariance fluxes at Bernal

Dominant flow at the Bernal site was from northeast (Figure 2). Energy balance closure for this site had a slope of 0.72 and $r^2$ = 0.92 (Figure 5). Homogeneous sites of the Fluxnet network obtain higher percentages of closure than 72%, and for the Bernal site, the vegetation heterogeneity was important (see below). Average H was always negative during nighttime, but during

some months of the dry and rainy season $\lambda$E was positive, particularly after down (Figure 6), as to allow nighttime evaporation from the soil or vegetation. However, during some months of the dry season rainfall was small (Figure 7), then the positive $\lambda$E suggested that cacti could have an active gas exchange at that time.

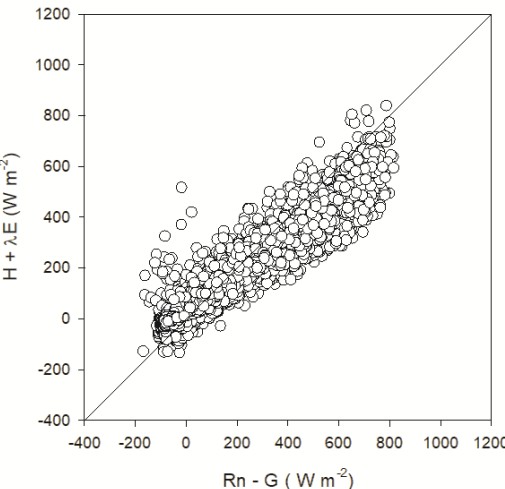

**Figure 5. Closure of the surface energy balance from eddy covariance measurements averaged at 30 min between the turbulent**
**fluxes (H+$\lambda$E) and available fluxes (Rn-G). Data is from march 30, 2017 to august 22, 2018 at Bernal site. The regression was y =**
**23.02 + 0.72 x, adjusted $r^2$ = 0.92. Diagonal line represents the 1:1 relation.**

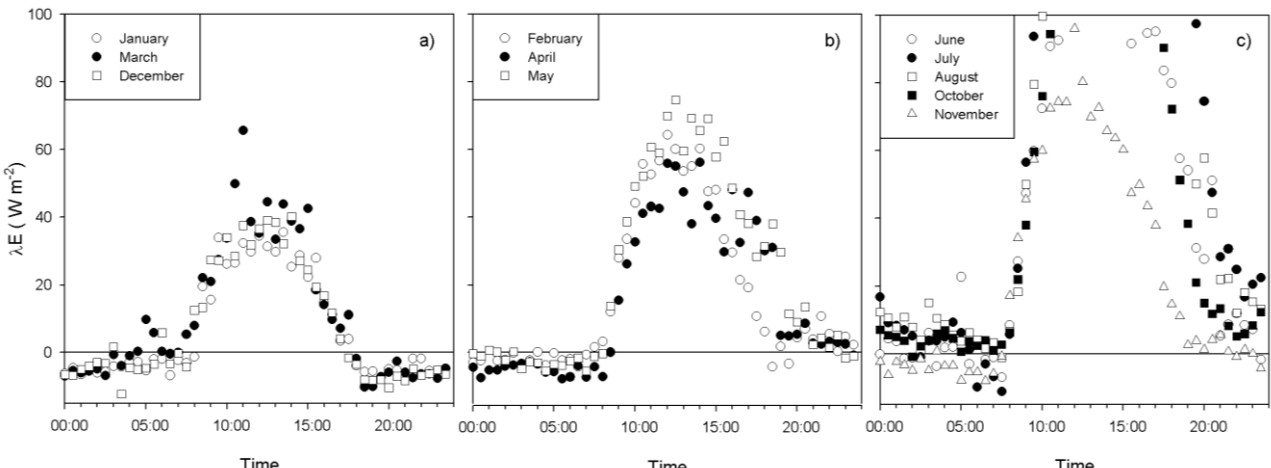

**Figure 6. Latent heat flux daily trend at Bernal during different months emphasizing nighttime λE. A) shows months with low rainfall in the previous month and predominantly negative λE during nighttime (these months had low rainfall: 0.17, 11.3, 0.33 mm rainfall for January, March and December). B) shows months with low rainfall in the previous month and positive λE after sunset (6.7, 7.3, 20.7 mm rainfall for February, April and May). C) shows months during the rainy season with λE positive mainly due to soil wetness and antecedent rainfall of 43, 202, 9, 190, 34 mm for June, July, August, October and November. Plot C) is out of scale in the y axis for compatibility with the other plots.**

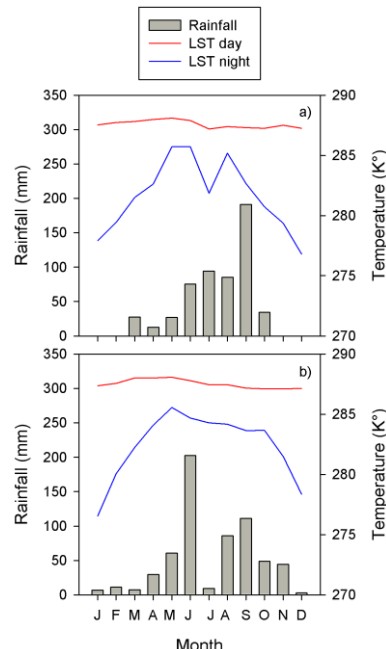

**Figure 7. Monthly rainfall and land surface temperature (LST) during years, a) 2017 and b) 2018 at Bernal site. The LST values correspond to the 1:30 PM (LST day) and 1:30 AM (LST night) MODIS Aqua satellite overpasses.**

Carbon dioxide absorptions had a diurnal behaviour beginning at dawn and ending before sunset (Figure 8). Nighttime flux was positive, indicating respiration, notwithstanding the presence of cacti. Although summer rains are characteristic of the climate at Bernal site (Figure 7), a negative NEE flux occurred at all measured months. The lowest $CO_2$ flux was recorded in January and February 2017 and in May 2018 (Table 1), this behaviour resulted from the phenology of the vegetation, since most species lost their leaves in the dry season, and also due to the effect of low temperature. Within the rainy season, the flux of $CO_2$ increased, compared to the months of January to June. The correlation between NEE and precipitation was -0.45. When the sum of the precipitation of the current month and that of the previous month was considered, the correlation with NEE was -0.7, suggesting that continuous availability of soil moisture is important for the absorption of $CO_2$ in this environment.

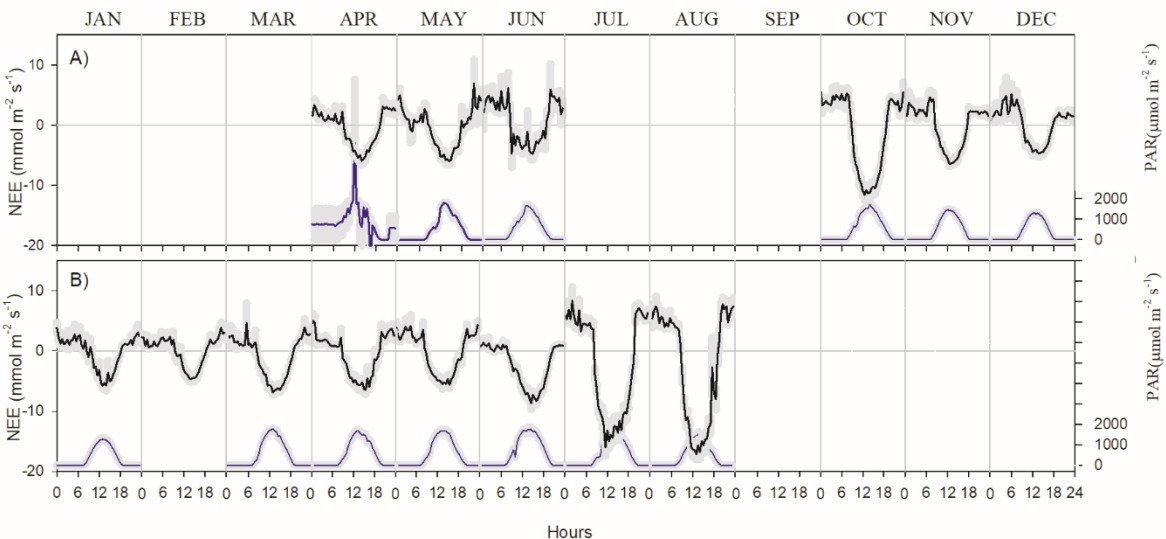

**Figure 8. Net ecosystem exchange (NEE) and photosynthetic active radiation (PAR) at Bernal site in A) 2017 and B) 2018. Negative values in the $CO_2$ flux indicate photosynthesis. The grey shadow is the standard error of mean for each month at any given hour.**

Table 1. Daily average values of the net ecosystem exchange (NEE), gross primary productivity (GPP), ecosystem respiration (R$_{eco}$) in a scrub at Bernal site. Negative values of NEE indicate photosynthetic absorption.

| | | NEE | GPP | R$_{eco}$ |
|---|---|---|---|---|
| | | | µmol m$^{-2}$ s$^{-1}$ | |
| **2017** | JAN | | | |
| | FEB | | | |
| | MAR | | | |
| | APR | -0.54 | 2.48 | 1.94 |
| | MAY | 0.05 | 2.86 | 2.91 |
| | JUN | 0.38 | 4.21 | 4.59 |
| | JUL | -1.00 | 1.78 | 0.77 |
| | AGO | | | |
| | SEP | | | |
| | OCT | -1.26 | 5.25 | 3.99 |
| | NOV | -0.13 | 2.65 | 2.52 |
| | DEC | 0.04 | 1.80 | 1.84 |
| **2018** | JAN | -0.05 | 1.29 | 1.24 |
| | FEB | 0.06 | 1.88 | 1.94 |
| | MAR | -0.94 | 2.45 | 1.51 |
| | APR | -0.58 | 2.87 | 2.29 |
| | MAY | -0.29 | 3.77 | 3.48 |
| | JUN | -2.52 | 4.33 | 1.81 |
| | JUL | -2.83 | 8.23 | 5.41 |
| | AGO | -1.93 | 9.21 | 7.28 |

The scrub at Bernal was heterogeneous in botanical composition. Twenty-four species of cacti and shrub were identified; on average, each sampling plot had 10.3 species. The IVI was similar between all cacti ($0.36 \pm 0.04$), shrub legumes ($0.38 \pm 0.04$) and other shrubs ($0.23 \pm 0.06$) sampled. Most sampling plots were at areas of high flux frequency (Figure 2). *Cylindropuntia imbricata* had the largest IVI, followed by *Acacia farnesiana, Acacia schaffneri* and *Prosopis laevigata*. The IVI of the herbaceous stratum represented by grasses was not characterized, due to the state of overgrazing and the absence of reproductive structures in plants, which made measurement difficult of their abundances, frequencies and dominances. The grass genera present were *Melinis, Chloris, Cynodon* and *Cenchrus*, corresponding all to invasive C$_4$ tropical grasses. Scrub species of higher IVI had a similar LAI (1.2), although the magnitude of the LAI of *P. laevigata* stood out (Table 2).

Table 2. Importance value index (IVI) and leaf area index (LAI) of the main species present at the Bernal, Querétaro study site.

| Species | Plant type | IVI | SEM[1] | LAI | SEM |
|---|---|---|---|---|---|
| *Coryphantha cornifera* | Cactus | 0.07 | 0.27 | | |
| *Bouvardia ternifolia* | Herb | 0.07 | 0.27 | | |
| *Karwinskia humboldtiana* | Shrub | 0.07 | 0.27 | | |
| *Forestiera phillyreoides* | Shrub | 0.09 | 0.27 | | |
| *Ferocactus latispinus* | Cactus | 0.09 | 0.27 | | |
| *Cylindropuntia leptocaulis* | Cactus | 0.09 | 0.27 | | |
| *Asphodelus fistulosus* | Shrub | 0.09 | 0.19 | | |
| *Brickellia veronicifolia* | Shrub | 0.10 | 0.27 | | |
| *Dalea lutea* | Shrub | 0.11 | 0.15 | | |
| *Eysenhardtia polystachya* | Legume | 0.13 | 0.15 | | |
| *Myrtillocactus geometrizans* | Cactus | 0.14 | 0.27 | | |
| *Schinus molle* | Shrub | 0.14 | 0.19 | | |
| *Jatropha dioica* | Herb | 0.15 | 0.19 | | |
| *Mammillaria uncinata* | Cactus | 0.16 | 0.12 | | |
| *Opuntia tomentosa* | Cactus | 0.17 | 0.11 | | |
| *Opuntia robusta* | Cactus | 0.23 | 0.07 | | |
| *Opuntia hyptiacantha* | Cactus | 0.26 | 0.07 | | |
| *Mimosa monancistra* | Legume | 0.28 | 0.12 | | |
| *Mimosa depauperata* | Legume | 0.31 | 0.12 | | |
| *Zaluzania augusta* | Shrub | 0.33 | 0.10 | | |
| *Viguiera linearis* | Herb | 0.36 | 0.11 | | |
| *Acacia schaffneri* | Legume | 0.41 | 0.07 | 1.13 | 0.15 |
| *Prosopis laevigata* | Legume | 0.41 | 0.07 | 1.48 | 0.12 |
| *Acacia farnesiana* | Legume | 0.56 | 0.09 | 1.12 | 0.37 |
| *Cylindropuntia imbricata* | Cactus | 0.74 | 0.07 | 1.13 | 0.11 |

[1] SEM: standard error of the mean.

## 3.2 Machine learning ensembles as predictors of eddy covariance GPP

In this section we describe the modeling with EML using local remotely sensed data from Bernal site to predict GPP at the same site, and then the agreement between EML GPP predictions and EC derived GPP. The automl function generated 1031 models with an average deviance of 1.35 while the deviance of the leader model was 0.63 in the training data set (Table 3). Eleven models of type GBM and five models XGBoost were in the top 30 models, along with nine best of family ensembles and five all models ensembles. The weighted variable importance in the leaderboard was higher for the LAI from MOD15 and MCD15 products (17% and 14%). The PsnNet, EVI (MOD17), FPAR (MCD15), the green atmospherically resistance vegetation index (GARI) and MODIS reflectance band 13, had an importance higher than 3% (5.9, 5.4, 4.2, 3.6 and 3.0%). LAI (MCD13 and MOD13), PsnNet and the FPAR (MOD15) were the more important variables (20, 17, 13 and 10%) in the top non-staked model, a GBM model that was ranked in fourth place.

Predictions of GPP in the testing dataset showed dispersion in the lower range of the scale of measurement and the correlation was 0.94 (Figure 9A). The final prediction of GPP for the whole dataset had a probability of agreement ($\theta$s) of $0.58 \pm 0.01$ (parameter estimate and standard error), $\alpha = -0.0616 \pm 0.11$ and $\beta = 1.0133 \pm 0.02$, suggesting a good fit with low fixed and proportional bias (Figure 9B). The probability of agreement decreased slightly at the lower and upper range of the scale of measurement (Figure 9C), indicating that the EML model would predict GPP without increasing the bias, particularly in the range from 0 to 4 g C $m^{-2}$ $d^{-1}$. However, the value of $\theta$s should be higher than 0.95 as to consider EC measurements and EML as interchangeable. The correlation of 0.99 ($r^2 = 0.98$) for the data in figure 9B could be misleading as it would suggest a very good fit.

Using only the five of the more important variables named above, to generate an EML, resulted in a XBoost leader model with 2.73 deviance and a total of 1094 models. Top 30 models were 16 XBoost and GBM models, and the best of family ensembles started to show up at the 12th place. Although the number of runs was the same (20), the automl function increased the number of produced models; but the smaller set of explanatory variables constrained the ability to identify features contributing to better models. Using another set of five randomly selected explanatory variables (one vegetation index and four MODIS bands) resulted in a leader model with 2.52 deviance out of 1024 models, but this time the leader was the best of the family ensemble. Using only a few variables was considered to increase the deviance compared with the average deviance of 1.35 obtained during the training phase and using all available variables.

Table 3. Leaderboard of EML models for the Bernal training dataset consisting of 85% of day observations. NA denotes the outcome where the type of model was not present in the 30 top performing models, according to deviance.

| Type of Model | Number of models | Average deviance | | |
| --- | --- | --- | --- | --- |
| | | All models | Top models | Leader model |
| Stacked Ensemble | | | | |
| All models | 20 | 0.852 | 0.834 | |
| Best of family | 20 | 0.812 | 0.703 | 0.633 |
| GBM | 453 | 1.366 | 0.796 | |
| DRF | 20 | 1.108 | NA | |
| XGBoost | 277 | 1.385 | 0.778 | |
| XRT | 20 | 1.077 | NA | |
| Deep Learning | 201 | 2.072 | NA | |
| GLM | 20 | 1.474 | NA | |
| Total | 1031 | 1.356 | | |

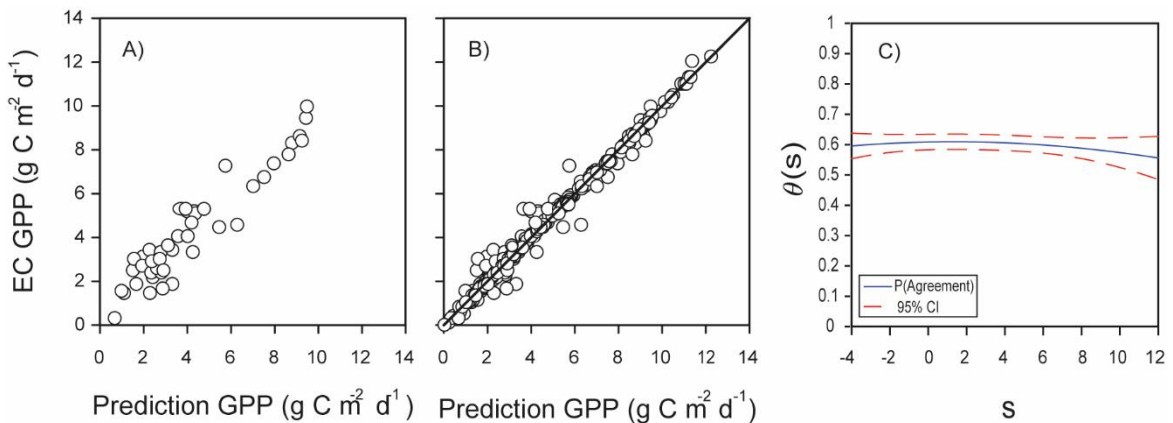

**Figure 9. Agreement between predictions of GPP obtained with machine learning algorithms or derived from eddy covariance measurements at Bernal site. A) Example of one run predictions of GPP in the test dataset from Bernal site using the leader model of an ensemble of machine learning algorithms (EML), the test dataset was 15% of data, r= 0.94. B) Predictions for the complete dataset, r= 0.99, the diagonal line is the 1:1 agreement. C) Function of probability of agreement using a 1 g C m⁻² d⁻¹ as tolerable agreement between methods of estimation of GPP corresponding to plot B). The horizontal axis (s) represents the magnitude of the measurement, the vertical axis is the probability of agreement for the measurement, and the red line is the confidence interval, p < 0.05.**

An important question for modeling upscaling is the capacity to extrapolate results temporally and spatially; here we explored the latter posing the following question: predictions of GPP from EML for a EC site would agree with EC observations from another site with "similar" environmental conditions? First an EML solution was found training 80% of the Santa Rita dataset obtaining a best of family ensemble with 0.23 deviance out of 634 trained models (here after this model is referred as the Santa Rita model). Then the environmental and remote sensed data from the Bernal site was fed into the Santa Rita model, this would be an external validation dataset. However, agreement was not good, the mean value of $\theta s$ was $0.15 \pm 0.01$, $\alpha = -1.0822 \pm 0.09$ and $\beta = 0.58127 \pm 0.02$. The value of $\theta s$ was not constant across the range of measurement and decreased rapidly after 2 g C $m^{-2}$ $d^{-1}$ (Figure 10B). Because the bias was important the predictions were adjusted using equation (6), showing some improvement with r = 0.78 (Figure 10A). Comparing figure 5B and 6A it is evident that an EML model extrapolation to other conditions is noisier, ie. Santa Rita model trying to represent the ecosystem function at Bernal. Notwithstanding, some of the most important variables were shared by both EML ensembles: Bernal and Santa Rita; in the case of Santa Rita LAI from MOD15, MYD15 and MCD15 had 35.0, 4.8 and 3.1% of variable importance and for the FPAR from MOD15 was 12%.

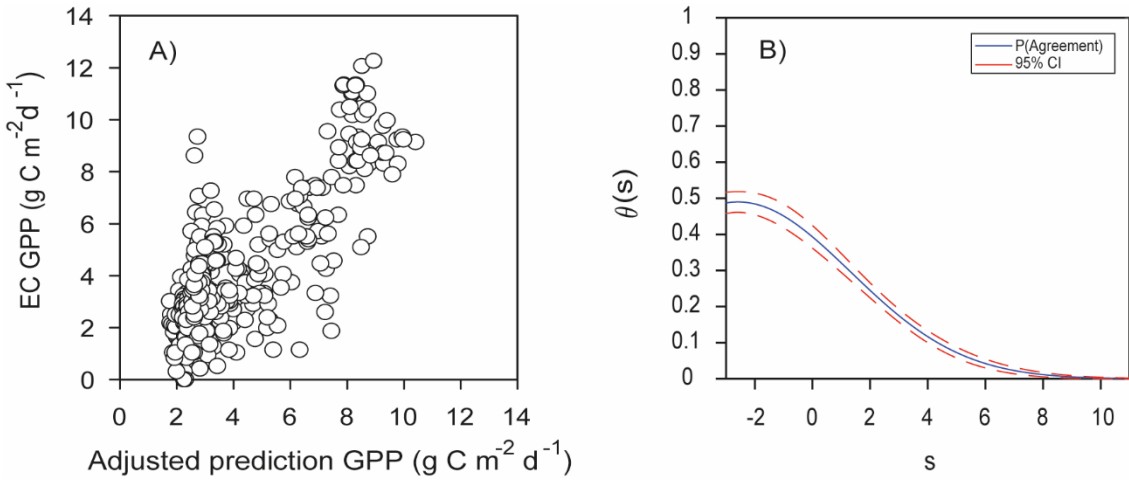

**Figure 10. A) Adjusted predictions of GPP for the complete dataset from Bernal site using the leader model of the final ensemble of machine learning algorithms (EML) derived from Santa Rita site compared to estimates of GPP from EC data, r= 0.78. B) Respective function of probability of agreement using a 1 g C $m^{-2}$ $d^{-1}$ as tolerable agreement between methods of estimation of GPP: EC data from the Bernal site and EML model for the Santa Rita site. The horizontal axis (s) represents the magnitude of the measurement, the vertical axis is the probability of agreement for the measurement, the red line is the confidence interval, p < 0.05.**

### 3.3 MODIS as predictor of eddy covariance GPP

MODIS is important because it overpasses every point of the earth every one or two days and it implements a GPP product (MOD17) that has helped track the response of the biosphere to the environment since 2000. The product MOD17 has been validated against many EC sites, but few validation sites correspond to deserts and semi-deserts (Running et al., 2004). The

GPP MOD17 underestimated the GPP derived from EC data at Bernal (Figure 11A). In a similar bell shaped distribution of θs, as in the case of the extrapolation of the Santa Rita site (Figure 10B), here the θs was not constant across the range of measurement; mean θs was $0.24 \pm 0.13$, $\alpha = 0.00047 \pm 0.087$ and $\beta = 0.48749 \pm 0.02$ (Figure 11C). Adjusting MOD17 estimates with equation (6) improved the relationship, but note that the value of r (0.76) was the same for original MOD17 and the adjusted MOD17 (Figure 11B).

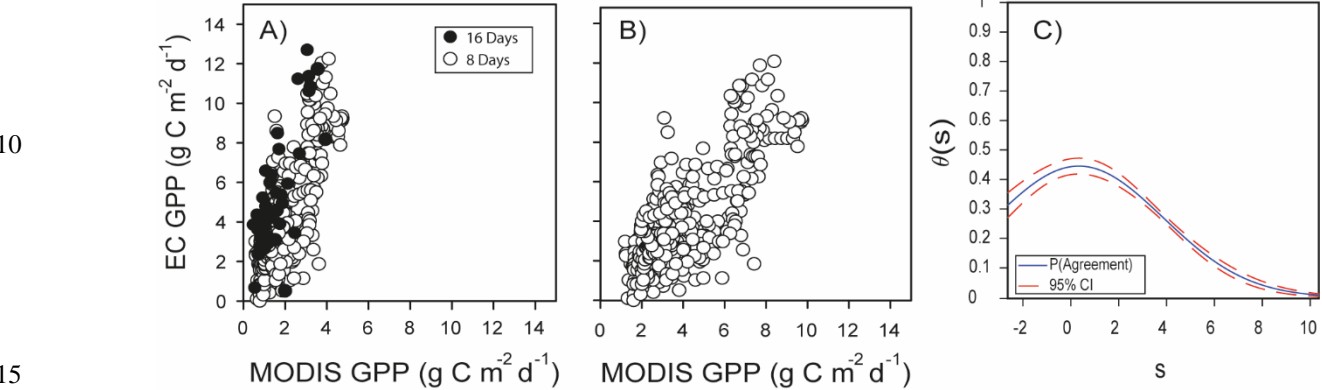

**Figure 11. A) MODIS17 estimates of GPP from Bernal site versus estimates of GPP from EC data using the complete dataset at a daily time step derived from spline for MOD17 GPP (○), r= 0.76 or, eight-day composite estimates obtained by eight-day averages of EC GPP (●), r= 0.76. B) Adjusted predictions of GPP from MODIS. C) Corresponding function of probability of agreement for Figure 7A using a 1 g C m$^{-2}$ d$^{-1}$ as tolerable agreement between methods of estimation of GPP: EC data from the Bernal site and MODIS MOD17. The horizontal axis (s) represents the magnitude of the measurement. The vertical axis is the probability of agreement for the measurement.**

### 3.4 Prediction eddy covariance GPP with ordinary least squares multiple regression

OLS is a common estimation method for linear models and here this model appeared as adequate, judging by the general distribution of predictions (Figure 12A) and the probability of agreement plot (Figure 12B). Fourteen variables were included in the model, all of them with VIF values lower than 7.0 (Appendix C); the VIF statistic quantifies the severity of multicollinearity and an acceptable threshold is 10. The most significant variables were the EVI from MYD13 and Daymet variables precipitation, short wave radiation, and minimum and maximum temperatures (see Appendix B for variable details). Variables with high coefficient values were MODIS reflectance band 14 (9.17) the EVI from MYD13 (8.53), and the NDVI (3.9), while Daymet temperatures had small coefficients: -0.23 for maximum temperature and 0.17 for minimum temperature. The θs decreased slightly at the ends of the measurement range; mean θs was $0.5 \pm 0.014$, $\alpha = 0.18845 \pm 0.137$ and $\beta = 0.94966 \pm 0.031$. No correction for this model was calculated since α was close to 0, β to 1 and the EML model had a higher θs.

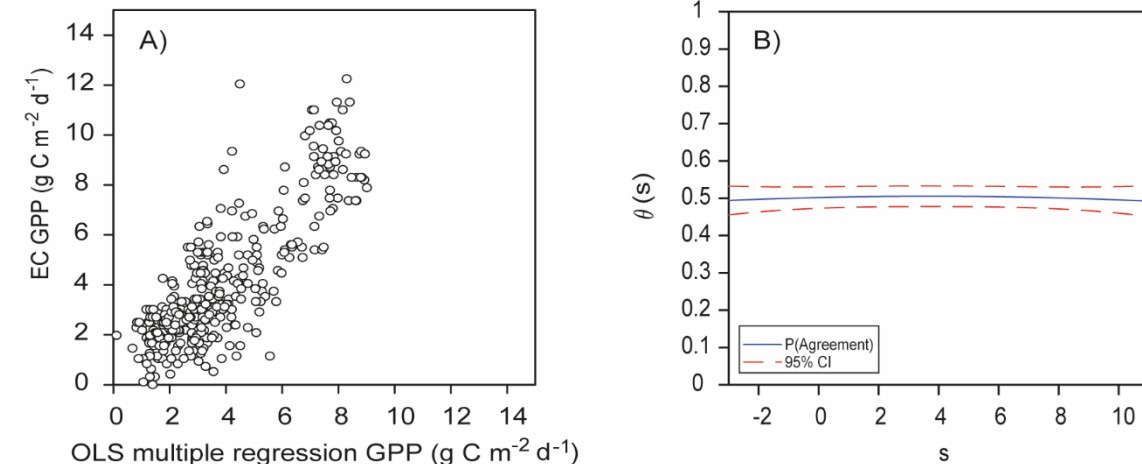

**Figure 12. A) Ordinary least squares multiple regression estimates of GPP for the complete dataset from Bernal site versus estimates of GPP from EC data, r= 0.82. B) Respective function of probability of agreement using a 1 g C m$^{-2}$ d$^{-1}$ as tolerable agreement between methods of estimation of GPP: EC data from the Bernal site and OLS multiple regression. The horizontal axis (s) represents the magnitude of the measurement. The vertical axis is the probability of agreement for the measurement.**

## 3.5 Agreement

The probability of agreement ($\theta$s) was the statistic to determine if the mean responses were in agreement. Comparing the confidence intervals (CI) for $\theta$s, the best modeling approach was the EML because its CI (0.56-0.59) was different from that of the OLS model (0.47-0.52). More importantly, for the EML and OLS models the values of $\theta$s had little variation across the range of GPP estimates. However, the CI of the EML and OLS models were similar in their bias estimates $\alpha$ and $\beta$ and both models had no bias (p <0.05). Altogether, the best result was the EML ensemble using environmental and remote sensed data corresponding to the same site, i.e. Bernal (Figure 10C). This kind of EML would be useful for gap filling or the evaluation of GPP time series of the site that generated the model. Machine learning algorithms can fill gaps longer than 30 days (Kang et al., 2019).

The second option to estimate GPP, using the same datasets, was the multiple regression OLS model (Figure 12A). The multiple regression is straightforward and here multicollinearity was not a problem. The EML ensemble and the OLS regression have the highest values of $\theta$s (0.58 and 0.5, respectively, Figure12B). Higher values for $\theta$s are desirable (>0.95) and this could be achieved by increasing the sample size, relaxing the tolerable magnitude for agreement (here was set at 1 g C m$^{-2}$ d$^{-1}$), or perhaps using different forcing variables.

The MODIS estimates were a third best alternative, since the mean $\theta$s was 0.24. In the present study we used a spline to fill the data to a daily time step since the MCD17 is an eighth day composite product; but a similar result was obtained if the EC GPP was rescaled and compared to the original eight-day MCD17 data (Figure 11A). The GPP from MODIS was an underestimate of EC GPP and when the estimates were adjusted (equation 6, Figure 11B) the performance was not better than

the OLS (model not shown). The MODIS land cover classification represented this site as grassland (MCD12) and this could be another reason for the poor agreement, besides the assumptions made in the MODIS algorithm regarding the $\varepsilon$ and $\varepsilon_{max}$ parameters and the response of vegetation to VPD. Agreement of MODIS GPP is crucial because MODIS products are frequently used in country wide assessments of the carbon cycle and can influence public policies.

The model with least agreement resulted when the EML ensemble generated from the Santa Rita site was used to predict GPP at Bernal. Machine learning models can make predictions but their usefulness decreases when they are used outside the context where they were built; while process based mechanistic models have this ability. Although the Automl function in the H2O is designed to protect against overfitting using cross validation runs (Michailidis, 2018), in our study, the Santa Rita model could not be generalized to represent the Bernal site GPP process. Probably because the variables and features selected for Bernal or

Santa Rita were different during the Automl workflow.

    The Santa Rita model was good at predicting GPP at that site, with a deviance of the leader model of 0.23 while at Bernal the deviance of the leader model was 0.63 (Table 3), indicating that the Santa Rita EML ensemble was at least as good model as the EML at Bernal (not shown). The GPP time series for Santa Rita was about four times the size of the Bernal dataset and therefore the deviance was lower. However, when the Santa Rita model was used with Bernal data the mean $\theta s$ was 0.16,

indicating that the agreement was insufficient. Eyeballing the predictions in Figures 9A, 10B and 11A and their corresponding correlation values (0.78, 0.76 and 0.82 for adjusted EML Santa Rita, adjusted MODIS and multiple regression OLS) it could be argued that these models were comparable. However, their $\theta s$ plots present a different perspective.

**4 Discussion**

    **4.1 Model agreement**

    The Santa Rita model had higher $\theta s$ when it extrapolated for low GGP values for the Bernal site, suggesting that the Santa Rita model had a better skill when predicting GPP close to zero and even negative GPP values (Figure 10B). Although the Bernal and Santa Rita sites had similar vegetation and climate classifications, they are more than 1600 km apart and rainfall monthly

distribution is different. The GPP seasonal cycle at Bernal started in February and steadily increased to a maximum during July and August (Table 1). At Santa Rita, the GPP was low from January until mid-July ($<0.5$ g $CO_2$ m$^{-2}$ d$^{-1}$) and then increased sharply to maximum level in mid-August (Joiner and Yoshida, 2020). What matters eventually for machine learning methods is how well the predictor space, rather than geographic space, is sampled (Jung et al., 2019). Incorporating data from more humid (semi-arid) sites could improve the GPP predictions by a machine learning method. Not only more EC sites, but also

sites representing a water availability gradient would be important for semi-arid ecosystems, representing long spells and the influence of oceanic oscillations and monsoon rains.

    All the models presented here used transient data to represent GPP, specifically, at the one-day time step. Besides the radiation related variables, two sources of rainfall were used as forcing variables but evapotranspiration (ET) was not used. The MOD16A2 version 6 is available as an 8-day gapfilled product and could be included in EML or OLS models. However, a

more general representation of the carbon cycling could be achieved when including variables that represent annual or seasonal timescales of soil water, evaporation or precipitation (Scott and Biederman, 2019). Scott et al. (2015) suggest that real lags between precipitation and productivity that may impart legacy effects may also be partially masked by using ET; as ET more carefully tracks productivity when soil moisture storage is accessed. The occurrence of off-season rainfall, dry spells and carry over effects could be parameterized as windowed events of a given duration. A window would be a period with distinct time boundaries, the window allows grouping records with similar features. The effect of the window at a given time could be represented as moving weights as the point in time in question is closer or farther from the window.

In our study the metric to assess model agreement was the probability of agreement ($\theta$s) and their bias parameters. Many other metrics can be used to evaluate model performance such as the root mean square error, r, $r^2$ or the model efficiency factor (MEF) presented by Nash & Sutcliffe (1970). In particular, the MEF is a step in the Fluxnet data processing pipeline (Pastorello et al., 2020). A MEF value close to 1 represents a high correlation and lower biases (Joiner and Yoshida, 2020). Therefore, the calculated MEF for the Bernal site EML (0.98) would suggest very good predictive performance, while the $\theta$s of 0.6 for this model indicates a more modest performance. The $\theta$s for a particular value of the measurand, the probability that the difference between two measurements made by different systems falls within an interval that is deemed to be acceptable (Stevens et al., 2015). In the present study we used 1 g C m$^{-2}$ d$^{-1}$ as a critical value defining an acceptable difference, if this value is smaller then, the probability of agreement would decrease for this same model. In such a case it would be less likely that the predictors agree considering that the $\theta$s takes into account both the difference in the function means at a given value of the measurand as well as the uncertainty in its estimation.

**4.2 MODIS discrepancies**

Different authors have reported discrepancies between MCD17 and EC estimates of GPP in semi-arid regions; examining MODIS discrepancies in these ecosystems is important because the errors induced by cloud cover are expected to be minimal and other effects can be identified (Gebremichael and Barros, 2006). The GPP of MOD17 did not relate well (EC = 0.11 + 0.17 MODIS, $r^2 = 0.67$) with estimates of EC GPP in semi-desert vegetation of Sahel (Tagesson et al., 2017). With data from different types of vegetation in the Heihe basin in China, MODIS17 overestimated the GPP from EC (EC = 1.15 + 0.24 MODIS, $r^2 = 0.68$, Cui et al., 2016). For scrub sites in Mexico, the relation between GPP calculated from EC and MOD17 was not good (MODIS = 383.82 + 0.467 EC, $r^2 = 0.6$, Delgado-Balbuena et al., 2018). In arid and semi-arid ecosystems in China, optimizing parameters of the MODIS GPP model with site-specific data, improved the estimate to explain 91% of the variation in the GPP of the data observed by EC (Wang et al., 2019). These same authors propose improving the land use classification used by the MOD17 algorithm and recalibrating light use efficiency parameters to solve the GPP estimation problem. Gebremichael and Barros (2006) examined an open shrubland site in a semi-arid region of Sonora, Mexico and their analysis of the temporal evolution of the discrepancies with MODIS GPP suggested revisiting the light use efficiency parameterization, especially the functional dependence on VPD and PAR and water stress or soil moisture availability.

The relationship between the GPP MODIS and the GPP EC presented in section 3.3 is an approximation, because the uncertainty in the respiration component must be considered. The empirical relationship between nocturnal NEE and soil temperature has been used to represent ecosystem respiration ($R_{eco}$) in order to separate the processes that contribute to daytime NEE (Richardson and Hollinger, 2005; Wofsy et al., 1993). Nighttime NEE should be equal to the rates of autotrophic and heterotrophic respiration, while during daytime, NEE should be equal to the combined rates of carboxylation and oxidation of RUBISCO, autotrophic respiration and heterotrophic respiration. Then the GPP can be calculated as the difference between daytime NEE and $R_{eco}$, estimated through its relationship with temperature (Goulden et al., 1996). In the present study, $R_{eco}$ was calculated based on soil/air temperature following the procedure of Reichstein et al. (2005) implemented in Reddyproc (Wutzler et al., 2018). Although it is possible to measure or model the partition of respiration (Running et al., 2004; Wang et al., 2018), the presence of cacti complicates the calculation, assuming that all nighttime flux represents ecosystem respiration (Owen et al., 2016; Richardson and Hollinger, 2005). While soil respiration tends to be temperature-limited when soil moisture is non-limiting in temperate ecosystems, in rangeland ecosystems the controls of soil $CO_2$ efflux were photosynthesis, soil temperature and moisture (Roby et al., 2019). In our study, the instrumentation did not include measurements of plant or soil respiration partition to validate the $R_{eco}$ estimates.

A problem regarding data comparison from remote orbital sensors and terrestrial observations is that different quantities are fundamentally measured. MODIS measures the radiation reflected by the earth's surface in two spectral bands at 250 m spatial resolution per pixel, five bands at 500 m and 29 bands at 1000 m. The EC technique has a footprint to measure $CO_2$ that varies dynamically in shape and size, but is generally considered to be 1 km², in this study 600 m. To solve the scaling, MODIS products related to the carbon cycle have been validated with the EC technique and biometric measurements on several spatial scales using process-based ecosystem models and characterizing areas up to 47 km$^2$ around the EC measuring tower (Cohen et al., 2003).

At Bernal the vegetation was heterogeneous, this situation was represented by the heterogeneity in vegetation activity in the four pixels used; ranging about 0.2 units of NDVI at the peak of the season activity (Figure 1). The standard error of the IVI differed by one order of magnitude among species. Although more important species had a lower standard error; their means were similar, indicating that they were equally abundant at all sampling plots. The higher importance of some species (Table 2) was explained by selective grazing-browsing behaviour and the dispersion caused by cattle, either by ingesting or transporting seeds or plant parts (Belayneh and Tessema, 2017). Regarding landscape heterogeneity, the tower fetch was predominantly from the northeast, capturing the less heterogeneous area of the site but also the more active, according to the NDVI (Figure 1).

The thorny scrub examined had two vegetation layers: the overstory layer mainly consisting of mesquite, acacia and cacti, and the understory layer that included grasses and herbs. Cattle preferentially graze the understory and because they eat using their tongue they will avoid browsing thorny species, unlike goats or deer that use their lips. Without grazing management, overtime, the competition balance will favour bush species resulting in encroaching and the understory will be stressed; only unpalatable species or those with their growing meristems very close to the ground would survive. Representing the structure and

functioning of these two layers using MODIS is possible (Liu et al., 2017). Recently, Hill and Guerschman (2020) presented a MODIS product derived from MCD43A4 to estimate the fractions of photosynthetic and non-photosynthetic vegetation and the remaining fraction of bare soil. These developments could improve the MOD17 GPP estimates, since its model represents a homogeneous single vegetation layer. All these considerations help to understand the low θs between MOD17 estimates of

GPP and EC derived GPP.

### 4.3 Forcing variables and machine learning

Forcing variables in this study were gridded meteorological data and MODIS spectral bands and products, but the variables mostly included in the ML algorithms were LAI and FPAR accounting for 36.4% at Bernal and 54.9% at Santa Rita of the variable importance index in their leaderboard. This result supported the view that $CO_2$ fluxes can be represented by ML

algorithms exclusively using remotely sensed data (Tramontana et al., 2016, Joiner & Yoshida, 2020). Using neural networks, Joiner & Yoshida (2020) showed that with only satellite reflectances from MODIS and top-of-atmosphere PAR, it is possible to capture a large fraction of GPP variability. They argue that vegetation indices may reduce the information content of the underlying reflectances when compressing the information from two or more bands into a single index. In our study, reflectance bands were not often included in the ML models. This was most likely because MOD09GA and MODOCGA provide estimates

of surface reflectance uncorrected for the illumination and viewing geometry, while the MCD43 used in Joiner & Yoshida (2020) includes BRDF correction. By propagating the BRDF correction in MODIS processing pipeline the vegetation indices more likely would better relate to the GPP. Future modeling efforts should use benchmark scenarios according to sets of forcing variables already identified as useful.

### 4.4 Carbon flux

Although the carbon balance in ecosystems is influenced by different factors such as soil type and amount of nutrients, the relationship with soil temperature and humidity is particularly strong (Anderson-Teixeira et al., 2011; Hastings et al., 2005). How much of the rainwater the system can retain or lose has been described as the leakiness of the system (Guerschman et al., 2009). More than immediate incident rainfall, the available soil moisture and its redistribution are important in semi-arid ecosystems, including steamflow, preferential flow paths, hydraulic lift and others (Barron-Gafford et al., 2017). At Bernal,

when the sum of the precipitation of the current month and that of the previous month was considered, the correlation with NEE was -0.7, suggesting that continuous availability of soil moisture is important for the absorption of $CO_2$ in this environment. This result is consistent with other studies in which the relationship between the net productivity of the ecosystem (NEP) and precipitation is initially positive, but is levelled from 1000 to 1500 mm annually (Xu et al., 2014). The hydraulic redistribution of water from moist (deeper) to drier soils through plant roots tended to increase modeled annual ecosystem

uptake of $CO_2$, this process was identified at US-SMR (Fu et al., 2018; Scott et al., 2003).

The leakiness is highly dependent upon vegetation fractional cover, the proportion of the surface occupied by bare soil and vegetation: photosynthetically active vegetation and non-photosynthetically active vegetation such as litter, wood and dead

biomass (Guerschman et al., 2009). All these vegetation fractions have a water storage capacity and can reduce the amount of effective rainfall available to plant roots. It is possible that the canopy of the bushes completely intercepted the rainfall in some months, because the scrub can intercept up to 20% of the precipitation and its canopy storage capacity is 0.97 mm (Mastachi-Loza et al., 2010). Considering only the daily rain events greater than 5 mm, the correlation between precipitation and NEE rose to -0.72. In the present study, the interception of rain by vegetation surfaces was not calculated, but the results suggest that it would be important to explore the relationship between net precipitation and NEE.

The average NEE at a global level is $-156 \pm 284$ g C m$^{-2}$ y$^{-1}$ (Baldocchi, 2014). The highest frequency among sites that measured NEE with EC occurs from -200 to -300 g C m$^{-2}$ y$^{-1}$, but in sites with biometric measurements, the peak occurs at -100 g C m$^{-2}$ y$^{-1}$ (Xu et al., 2014). Using the daily averages of Table 2, the average NEE during the measurement period was -0.78 g C m$^{-2}$ d$^{-1}$ and annually would be -283.5 g C m$^{-2}$ y$^{-1}$. This result was higher than the annual values of the induced grassland and scrubland vegetation characterizing the Sonora desert plains (138 and 130 g C m$^{-2}$ y$^{-1}$, Hinojo-Hinojo et al., 2019). In New Mexico, NEE values measured with EC are between 35-50 g C m$^{-2}$ y$^{-1}$ in desert grassland and 344-355 g C m$^{-2}$ y$^{-1}$ in mixed coniferous forest (Anderson-Teixeira et al., 2011). In a dryer region, the sarcocaulescent scrubland of Baja California in Mexico, the NEE was -39 and -52 g C m$^{-2}$ y$^{-1}$ in 2002 and 2003, respectively (Hastings et al., 2005). The NEE measured here was within the range of NNE $0.3 \pm 0.2$ kg C m$^{-2}$ yr$^{-1}$ for grasslands/shrublands in Mexico (Murray-Tortarolo et al., 2016). Although the measurements of the present study had gaps and were compared with annual studies, we considered that the reported value of C was representative of the main season of growth of this type of scrub.

**4.5 Ecosystem management**

Overgrazing is an appreciation relative to the grazing productive system where the forage resource is overused; in a mixed shrub-grass ecosystem, such as Bernal, usually refers to the understory. Overgrazing means that the plant regrowth is readily grazed, tillers and root reserves are lost and eventually the plant may die. Although the Bernal site was overgrazed, the carbon fluxes indicated that the plant community was photosynthetically active in both the dry and rainy seasons. It is fair to assume that the water in the soil was not limiting for the deep rooted bush species and that was the reason why it was possible to maintain the photosynthetic function during rainless months. However, this primary production would not have tangible benefits for rancher's production system, since no edible biomass would be produced for the cattle. From the point of view of carbon capture, the system accumulated non-labile biomass that would remain in the system for a longer time compared to a grassland ecosystem (although it would be necessary to determine the partition of said shrub biomass). However, the overgrazing condition affects the biomass of the understory roots and consequently the carbon pool in the soil.

In the short term, it can be thought that the estimated negative carbon flows are a favourable effect on the environmental agenda. As time passes, it is possible that the gaps between the individual shrubs of the overstory expand and this would have had an effect on soil erosion. It is also possible that the water stored in the soil profile used by the bushes gradually decreases, to the point of causing drought, changes in phenology and advancing the desertification process. There are many opportunities for ecology conservation and livestock-oriented management; this may include controlled grazing or propagating native

thornless shrub species. If the ranchers do not identify a benefit in the vegetation, then they will be tempted to remove it, as it occurred at the study site. Because of its wide coverage and readily availability the MODIS GPP product, accuracy is important in representing the carbon cycle, raising awareness and monitoring advancement of environmental decisions.

Although we found that the EML was a good option for modeling the GPP of a site, what is really needed to evaluate the performance of semiarid ecosystems is a spatial representation of the carbon flux. This is a problem for an underrepresented area regarding instrumented EC towers. However, the EML could be designed to take into account the explanatory variables in a spatiotemporal continuity. As demonstrated here, extrapolating the EML model from one site to another had poor agreement.

## 5 Conclusion

The best modeling approach was the ensemble of machine learning, the second option to estimate GPP was the multiple regression OLS and the third alternative were the MODIS estimates. Machine learning was a good option to predict GPP in the context were it was generated, otherwise its performance was not good. Nevertheless, a machine learning model would be useful for gap filling or the evaluation of GPP at the same site. The GPP estimates of a given model can be adjusted using the bias parameters of the probability of agreement to improve the relationship.

The Bernal site was a carbon sink notwithstanding its overgrazed condition. This is due to the contribution to the carbon flux of the predominating shrub species in this area. Although the importance value index of cacti was high in the study area, their metabolic activity did not outweigh the respiration component of the $CO_2$ flux during nighttime. Therefore, it is necessary to measure autotrophic and heterotrophic respiration components of the ecosystem as well as an alternative to $CO_2$ EC measurement during nighttime.

### Data availability

Database and programming code are available at https://doi.org/10.5281/zenodo.3598595

### Author contribution

Guevara-Escobar contributed to conceptualization, formal analysis, investigation, methodology, software, supervision, validation, visualization, writing – original draft and review & editing.

E. González-Sosa was in charge of funding acquisition, project administration and writing – review.

M. Cervantes-Jimenez contributed to investigation, methodology, validation, visualization, writing – original draft and review, editing and communication.

M. E. Queijeiro-Bolaños helped with investigation, validation, writing – review.

H. Suzán-Azpiri contributed with conceptualization and writing – review.

I. Carrillo- Ángeles helped with investigation and writing – review.

V. H. Cambrón-Sandoval contributed with writing – review and funding acquisition.

**Competing interest**

The authors declare that they have no conflict of interest.

**Acknowledgments**

We thank NASA and Ameriflux for data availability. The work was financially supported by CONACyT - SEMARNAT project no. 2014-1-(249407). Data filtering was carried out by the cluster Abacus I, Centre of Applied Mathematics and High Performance Computing ABACUS-CINVESTAV.

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

## Appendix A

Table A1. Threshold of velocity friction (u*) above which night-time fluxes were considered valid. Estimates were obtained using the R package Reedyproc. We used only the flux records with u* equal or higher than the corresponding mean u* threshold for each year. Bound values are a 95% confidence interval.

| Aggregation method | Year | Season | Mean | Lower bound 5% | Upper bound 95% |
|---|---|---|---|---|---|
| | | | u* (m s$^{-1}$) | | |
| Single | NA | NA | 0.193 | 0.140 | 0.209 |
| Year | 2017 | NA | 0.194 | 0.128 | 0.227 |
| Year | 2018 | NA | 0.193 | 0.132 | 0.225 |
| Season | 2017 | 2016012 | 0.194 | 0.128 | 0.227 |
| Season | 2017 | 2017003 | 0.215 | 0.141 | 0.297 |
| Season | 2017 | 2017006 | 0.194 | 0.128 | 0.227 |
| Season | 2017 | 2017009 | 0.181 | 0.144 | 0.208 |
| Season | 2018 | 2017012 | 0.149 | 0.116 | 0.232 |
| Season | 2018 | 2018003 | 0.271 | 0.134 | 0.297 |
| Season | 2018 | 2018006 | 0.193 | 0.132 | 0.221 |

Appendix B

Table B1. MODIS reflectance bands and products database.

| Product name | Satellite layer | Spatial resolution | Temporal resolution (days) | Spectral coverage (nm) |
|---|---|---|---|---|
| MOD09GA | Band1 | 500 m | 1 | 620-670 |
| | Band2 | | | 841-876 |
| | Band3 | | | 459-479 |
| | Band4 | | | 545-565 |
| | Band5 | | | 1230-1250 |
| | Band6 | | | 1628-1652 |
| | Band7 | | | 2105-2155 |
| MODOCGA | Band8 | 1 km | | 405-420 |
| | Band9 | | | 438-448 |
| | Band10 | | | 483-493 |
| | Band11 | | | 526-536 |
| | Band12 | | | 546-556 |
| | Band13 | | | 662-672 |
| | Band14 | | | 673-683 |
| MOD17A2H | Gross Primary Production (GPP) | | 8 | N/A |
| | Net Photosynthesis (PsnNet) | | | |
| MOD15A2H | Fraction of Photosynthetically Active Radiation (Fpar) | | | |
| | Leaf Area Index (LAI) | | | |
| MOD11A2 | Land Surface Temperature and Emissivity of day (LST day) | 1 km | | |
| | Land Surface Temperature and Emissivity of night (LST night) | | | |

Table B1. MODIS reflectance bands and products database. (Continuation)

| Product name | Satellite layer | Spatial resolution | Temporal resolution (days) | Spectral coverage (nm) |
|---|---|---|---|---|
| MYD13A1 | Enhanced Vegetation Index (EVI) | 500 m | 16 | N/A |
| MYD15A2H | Fraction of Photosynthetically Active Radiation (Fpar) <br> Leaf Area Index (LAI) | | 8 | |
| MYD11A2 | Land Surface Temperature and Emissivity of day (LST day) <br> Land Surface Temperature and Emissivity of night (LST night) | 1 km | | |
| MCD15A2H | Fraction of Photosynthetically Active Radiation (Fpar) <br> Leaf Area Index (LAI) | 500 m | | |

Table B2. Daily vegetation indexes computed using the MODIS reflectance bands described in Table 1.

| Index | Formula | Reference |
|---|---|---|
| Simple Ratio (SimpleR) | $SimpleR = \dfrac{Band2}{Band1}$ | |
| Moisture CStress (MoistS) | $MoistS = \dfrac{Band6}{Band2}$ | |
| Disease Stress Index (DSI) | $DSI = \dfrac{Band2 + Band4}{Band6 + Band1}$ | (Hill *et al.*, 2006) |
| Red Green Ratio Index (RGRI) | $RGRI = \dfrac{Band1}{Band4}$ | |
| Normalized Difference Vegetation Index (NDVI) | $NDVI = \dfrac{Band2 - Band1}{Band2 + Band1}$ | |
| Normalized Difference Water Index (NDVI_w) | $NDVI\_w = \dfrac{Band2 - Band5}{Band2 + Band5}$ | |
| Green Leaf Index (GLI) | $GLI = \dfrac{2 * Band11 - Band14 - Band9}{2 * Band11 + Band14 + Band}$ | |
| Green Atmospherically Resistance Vegetation (GARI) | $GARI = \dfrac{Band5 - (Band11 - (Band9 - Band14))}{Band5 - (Band11 + (Band9 - Band14))}$ | (Henrich *et al.*, 2012) |
| Enhanced Vegetation Index (EVI) | $EVI = 2.5 * \dfrac{Band2 - Band1}{Band2 + (6 * Band1) - (7.5 * Band9) + 1}$ | |

Table B3. Daymet meteorological database.

| Variable | Spatial resolution | Temporal resolution | Reference |
|---|---|---|---|
| Precipitation (Dayprc) | | | |
| Shortwave radiation (Daysrad) | | | |
| Maximum air temperature (DayTmax) | 1 km | Daily | (Thornton et al., 2017) |
| Minimum air temperature (DayTmin) | | | |
| Water vapor pressure (Dayvp) | | | |

Table B4. Precipitation data.

| Satellite | Product name | Spatial resolution | Temporal resolution |
|---|---|---|---|
| Global Precipitation Measurement (GPM) | 3IMERGDF v006 | 0.1 Degree | |
| | | | Daily |
| Tropical Rainfall Measuring Mission (TRMM) | 3B42 v007 | 0.25 Degree | |

Table C1. Analysis of variance for GGP derived from EC data at Bernal site. Variable details are described in Appendix B.

| Model term | Coefficient | EE[1] | DF[2] | SC[3] | F | p | VIF[4] |
|---|---|---|---|---|---|---|---|
| Regression | | | 14 | 1780.48 | | | |
| Constant | -13.18 | 6.48 | | | -2.03 | 0.043 | |
| R_08_405.420[5] | -2.119 | 0.602 | 1 | 26.8 | -3.52 | 0 | 2.2 |
| R_13_662.672[5] | -2.86 | 1.23 | 1 | 11.61 | -2.32 | 0.021 | 1.55 |
| R_14_673.683[5] | 9.17 | 2.59 | 1 | 27.06 | 3.53 | 0 | 1.22 |
| RGRI[6] | -0.832 | 0.369 | 1 | 11 | -2.25 | 0.025 | 1.43 |
| GARI[6] | 2.01 | 1.08 | 1 | 7.55 | 1.87 | 0.063 | 2.37 |
| EVIMYD[7] | 8.53 | 2.01 | 1 | 39.03 | 4.25 | 0 | 2.94 |
| NDVI_w[6] | 3.9 | 1.55 | 1 | 13.67 | 2.51 | 0.012 | 2.71 |
| PsnNet[7] | 0.01312 | 0.00223 | 1 | 75.3 | 5.9 | 0 | 3.83 |
| LstNgtMYD[7] | 0.0462 | 0.0238 | 1 | 8.18 | 1.94 | 0.053 | 2.01 |
| Dayprc[8] | 0.1388 | 0.0341 | 1 | 35.9 | 4.07 | 0 | 1.92 |
| Daysrad[8] | 0.0106 | 0.00218 | 1 | 51.21 | 4.86 | 0 | 5.01 |
| DayTmax[8] | -0.2386 | 0.0581 | 1 | 36.58 | -4.11 | 0 | 6.89 |
| DayTmin[8] | 0.1752 | 0.0454 | 1 | 32.22 | 3.86 | 0 | 4.96 |
| TRMM[9] | -0.0434 | 0.0194 | 1 | 10.87 | -2.24 | 0.026 | 1.41 |
| Error | | | 403 | 872.97 | | | |
| Total | | | 417 | 2653.45 | | | |

1 Standard error of coefficient.

2 Degrees of freedom.

3 Adjusted sum of squares.

4 Variance inflation factor.

5 Variable name denotes the band number and spectral bandwidth of MODIS (Moderate Resolution Imaging Spectroradiometer).

6 Vegetation indices RGRI is red green ratio index: and GARI is green atmospherically resistance vegetation index, details of formula are described in AppendixB.

7 Layers from MODIS products. EVIMYD is enhanced vegetation index from MYD13

PsnNet is photosynthesis product form MOD17, LstNgtMYD is nighttime land surface temperature emissivity from MYD11.

8 Variables obtained from Daymet daily dataset: DayTmin is minimum temperature, DayTmax is maximum temperature, Daysrad is shortwave radiation, Dayprc is precipitation.

4 Daily rainfall rate from 3B42 TRMM (Tropical Rainfall Measuring Mission).