# Peer review of "Machine learning estimates of eddy covariance carbon flux in a scrub in the Mexican highland"

_Biogeosciences, 2019_

## Referee Comment (RC1) · Anonymous Referee #1 · 16 Jan 2020

The manuscript by Guevara-Escobar and co-authors present a brief description of CO2 fluxes in a xerophytic shrubland in Mexico. The topic is relevant for Biogeosciences as much more information is needed in water limited ecosystems, underrepresented regions around the world, and ecosystems with different metabolic strategies (e.g., CAM, C3, C4). That said, I found the manuscript and the information presented limited in scope and premature.

Comments:

Introduction: The introduction lacks a clear scientific question and related hypothesis. Providing new measurements of NEE at underrepresented ecosystems is important, as well as comparing GPP estimates with satellite-derived products. That said, this manuscript should emphasize what is new (beyond new measurements) and have a

testable hypothesis if possible. The introductions resemble a technical report and could be improved by framing it around a clear scientific question.

Methods section: This section requires substantial reorganization and more information. The authors should link the methods to the research questions/hypotheses, provide more information about the site and how data was processed and analyzed. Maybe a section about data analysis would help to improve this section.

Results/Discussion section: I strongly recommend separating results from the discussion. Without a clear scientific question and testable hypothesis, it is difficult to evaluate this section and the novelty of the results. The authors touch different topics from leaf level photosynthesis, ecosystem level fluxes and remote sensing but I feel that there is disconnection between the results in this section. Finally, due to the limited dataset and analyzes, this section seems to be over-interpreting the results and consequently I wonder if this manuscript is premature for this study site.

Conclusion: I believe that this section is not fully supported by the data analysis and results. Again, it is difficult to evaluate this manuscript as the authors touch several topics, but none is analyzed in detail leaving the manuscript presenting a very broad (and potentially over interpreting) view of results. I respectfully believe that this is a good first step to summarize results from this study site, but this study requires substantial improvements in quality and quantity of data (e.g., longer datasets), conceptual organization (e.g., questions/hypotheses), and further analyzes to test clear hypotheses to provide a novel scientific contribution.

Figure 1 is difficult to interpret because the legend is not informative. I interpret it as mean diel patterns for the months represented in the figure, where the top panel is 2017 and the bottom one 2018. It is not clear why the authors present diel means and not the actual data or how many days were used to calculate the diel means for each day. Consequently, the methods section needs much more description about data quality, data availability, and data analysis to fully evaluate these results and the discussion.

Comments in detail

Study site: Description of the study site could be improved by following BADM guidelines for AmeriFlux. Although compiling all variables is challenging, a better description of the site is needed in order to compare this site with others across the world.

Lines 85-86 – I disagree with this statement as reporting energy balance closure is a good practice for data interpretation and data comparison.

Lines 87-95 – This belongs to data QA/QC and flux partitioning but a better description is needed.

Section 2.3 – Why not simply using the ORNL DAAC MODIS/VIIRS land product subsets tool?

Line 121-122 – LAI was only measured 2 times? Why only two days and not reporting a seasonal trend? Where those dates representative for maximum LAI?

Line 125 - It is unclear how the use of the Li-6400XT fits into the main purpose of the manuscript. How these data were used? Any upscaling approach?

Section 5 data availability: The proper place to host the eddy covariance data would be a standardize repository such as AmeriFlux or FLUXNET. Zenodo is a good place for overall code and ancillary datasets from this study but I appreciate the effort for archiving the dataset.

Figure 2 – How the eddy covariance data was aggregated for this analysis?

Figure 3- More discussion about why this figure is presented and what does it means for addressing a scientific question is needed.

---

## Author Comment (AC1) · 20 Feb 2020

We would like to thank the reviewer constructive comments and annotations. Please find our reply marked along with the reviewers text.

The manuscript by Guevara-Escobar and co-authors present a brief description of CO2 fluxes in a xerophytic shrubland in Mexico. The topic is relevant for Biogeosciences as much more information is needed in water limited ecosystems, underrepresented regions around the world, and ecosystems with different metabolic strategies (e.g., CAM, C3, C4). That said, I found the manuscript and the information presented limited in scope and premature.

Response: We agree with the reviewer impression on the limited scope of our study

because it only analyzed one site and roughly one year of eddy covariance (EC) data. Also, the data set is small compared with recent papers reporting advances on C flux. We intent to improve the paper with a more complete elaboration of the main question, which is the agreement between two methods of measurement: the EC and the MODIS algorithm. MODIS method is based on the radiation use efficiency logic to predict GPP (gross primary production). To support this approach, we propose to:

1) Calculate adequate parameters for the radiation use conversion efficiency (ïĄěmax), the photosyntetically active radiation that is absorbed for the vegetation (APAR) at our site (Bernal) following Running et al. (2000) and use the parameters from the BPLUT (Biome Properties Look-up Table) corresponding to the most similar vegetation type (Heinsch et al., 2003) to calculate eight-day GPP.

2) Use the available data layers from MODIS (Aqua and Terra) and generate a random forest regression ensemble for eight-day GPP (Tramontana et al. 2016). The prediction of GPP applying machine learning algorithms used different forcing variables, including those of MODIS terra (Tramontanta et al. 2016, Jung et al. 2019), but apparently not from MODIS aqua. Therefore, something interesting would be to test if both MODIS products contribute to a better representation of GPP using random forest. The rationale would be that the visit time of these sensors is different during the day and different QC scores can be expected.

3) Compare these GPP values against the GPP modeled from the EC using a Bayesian approach (Stevens et al. 2017), and perform a validation exercise with the best GPP estimate using a small dataset that was not included in the present paper.

Perhaps it will be more interesting to the reader when current methods are applied to obtain a better estimate of GPP using the MODIS platform. We look forward to publish this work even when the data is limited. Please consider the following reasons and the response to the specific comments. We decided to summit the manuscript with the available data for a number of reasons: a) The Bernal site suffered important

changes in land use during 2019 and scrub was suddenly cleared, we are not able to present a longer time series for this site. b) Real estate development, feedlot beef production, cheese and wine production associated with tourism, and automotive industry development are very attractive options for landowners in the region. c) Local society (including authorities) do not recognize scrub and shrubs as resource-valuable or as an ecosystem with a role for better livelihoods; compared to forests. d) Although the landowner agreed in principle and allowed a monitoring site, we can see that people need better information about environmental services. Otherwise, this vegetation type will remain very vulnerable to land use change.

We do not have any research station that would be desirable to assure research continuity. Selecting the Bernal site was a gamble, but getting a glance at an overgrazed site was important since there are few publications of C flux data from grazing conditions in general. The EC data showed that scrub at Bernal was a carbon sink. This estimate is not only important in science, but could have an impact on public policy and management. Only one year of EC data stresses the need for more work regarding number of sites and long term studies. Here, we presented only one year of data, but the assessment of interannual variability is desirable. We could develop an upscaling approach at Bernal to explore this aspect (as the reviewer suggests).

Mexico as many other countries have few C flux monitoring sites and remote sensed data is an important alternative to represent vegetation functioning, modeling and other applications. Relaying on remote data is partially cost-effective since most data from NASA is open access. Therefore, the main question in the present work was: estimates of the CO2 flux are comparable between ground-based and remote sensed methods? This question is not fully answered yet, since some publications shows that the relationship is not very good. Other reports indicate that the relationships are specific to vegetation types. Although Mexico has 30% of scrub and shrublands they differ in their botanical composition and structure, not mentioning management practices. A multi-site study would have interesting hypothesis to test, but that is beyond the capabilities

of our group. In Mexico there is a handful of sites representing scrub but these are very different in climate and plant species. Mexflux is a network supporting efforts for C flux research and data sharing arrangements are not complete. Lastly, Ameriflux, Fluxnet and other networks list few sites from Mexico.

Introduction: The introduction lacks a clear scientific question and related hypothesis. Providing new measurements of NEE at underrepresented ecosystems is important, as well as comparing GPP estimates with satellite-derived products. That said, this manuscript should emphasize what is new (beyond new measurements) and have a Discussion paper testable hypothesis if possible. The introductions resemble a technical report and could be improved by framing it around a clear scientific question.

Response: Background information required to better understand this paper will be included and the hypothesis will be explicitly framed to demonstrate that the vegetation at Bernal site was a sink of carbon and also to test the relationship between remote sensed GPP from MODIS and measurements from the EC tower. We will include in this section information about the importance of new measures in scrub ecosystems. The introduction includes elaboration about the world-wide relevance of MODIS products, the background of the MODIS16 algorithm and its upscaling validation. However, upscaling $CO_2$ fluxes (GPP or NEE) from eddy covariance sites could have different approaches as proposed in FLUXCOM initiative using machine-learning methods, basically using MODIS data along with meteorological data. Following Tramontana et al. (2016), a range of forcing variables from MODIS will be used with a random forest algorithm to upscale data from Bernal site. A small dataset from a period of 2018, not included in the present work, could be used for validation. To support this analysis, the introduction will include a presentation of the radiation use efficiency process-based algorithm as used in the MODIS MOD17 products and an alternative bottom-up machine learning algorithm. Upscaling fluxes using this approach should have better performance in comparison to the obtained results using simple Theil Sen regression.

Methods: This section requires substantial reorganization and more information. The

authors should link the methods to the research questions/hypotheses, provide more information about the site and how data was processed and analyzed. Maybe a section about data analysis would help to improve this section. The section will be reorganized and will include time series of phenology and climatic variables.

Response: Recommendations will be followed accordingly to explain data analysis.

Results/Discussion section: I strongly recommend separating results from the discussion. Without a clear scientific question and testable hypothesis, it is difficult to evaluate this section and the novelty of the results. The authors touch different topics from leaf level photosynthesis, ecosystem level fluxes and remote sensing but I feel that there is disconnection between the results in this section. Finally, due to the limited dataset and analyzes, this section seems to be over-interpreting the results and consequently I wonder if this manuscript is premature for this study site.

Response: We will separate results and discussion in two sections. All the information related to the site description will be incorporated to the methods section, i.e. leaf level photosynthesis and vegetation structure. This section will focus on presenting the time series of CO2 and the relation between EC data and upscaled estimates.

Conclusion: I believe that this section is not fully supported by the data analysis and results. Again, it is difficult to evaluate this manuscript as the authors touch several topics, but none is analyzed in detail leaving the manuscript presenting a very broad (and potentially over interpreting) view of results. I respectfully believe that this is a good first step to summarize results from this study site, but this study requires substantial improvements in quality and quantity of data (e.g., longer datasets), conceptual organization (e.g., questions/hypotheses), and further analyzes to test clear hypotheses to provide a novel scientific contribution.

Response: The conclusion will use only the evidence presented.

Figure 1 is difficult to interpret because the legend is not informative. I interpret it as

mean diel patterns for the months represented in the figure, where the top panel is 2017 and the bottom one 2018. It is not clear why the authors present diel means and not the actual data or how many days were used to calculate the diel means for each day. Consequently, the methods section needs much more description about data quality, data availability, and data analysis to fully evaluate these results and the discussion.

Response: Figure legends will be modified. In the methods we will describe the details of data quality and availably as requested. The reviewer is correct this figure presents data corresponding to two years. values are diel means and standard errors.

Comments in detail

Study site: Description of the study site could be improved by following BADM guidelines for AmeriFlux. Although compiling all variables is challenging, a better description of the site is needed in order to compare this site with others across the world.

Response: We will make the best effort prepare the description of the site following the BADM templates.

Lines 85-86 – I disagree with this statement as reporting energy balance closure is a good practice for data interpretation and data comparison.

Response: We will present the energy balance as recommended.

Lines 87-95 – This belongs to data QA/QC and flux partitioning but a better description is needed.

Response: This will be reorganized as suggested.

Section 2.3 – Why not simply using the ORNL DAAC MODIS/VIIRS land product subsets tool?

Response: The subsets tool is very useful and data retrieval is fast but the QC layer is not included.

Line 121-122 – LAI was only measured 2 times? Why only two days and not reporting a seasonal trend? Where those dates representative for maximum LAI? Line 125 - It is unclear how the use of the Li-6400XT fits into the main purpose of the manuscript. How these data were used? Any upscaling approach?

Response: Time trends of MODIS LAI, NDVI and EVI will be included. Measurements of LAI and photosynthesis were mentioned briefly to back up the site description, but they were not used to support any hypothesis.

Section 5 data availability: The proper place to host the eddy covariance data would be a standardize repository such as AmeriFlux or FLUXNET. Zenodo is a good place for overall code and ancillary datasets from this study but I appreciate the effort for archiving the dataset.

Response: Half-hourly flux data and tower metadata will be uploaded to Ameriflux. Code and data used for figures will be available at Zenodo.

Figure 2 - How the eddy covariance data was aggregated for this analysis?

Response: The results of ReddyProc were averaged every eight days according to the MODIS MOD17A2 timestamp. Although in the methodology we will explain it, the figure caption will include this information. Thanks for pointing this out.

Figure 3- More discussion about why this figure is presented and what does it means for addressing a scientific question is needed.

Response: We present the standard deviation of the MODIS data to indicate dispersion presented by the sensor data, especially in the rainy season, which helps to think that the adjustment should be made in two seasons.

References Heinsch FA, Reeves MC, Votava P, Kang S, Milesi C, Zhao M, Glassy J, Jolly WM, Loehman R, Bowker CF, Kimball JS, Nemani RR, Running SW. 2003. User's Guide: GPP and NPP (MOD17A2/A3) Products, NASA MODIS Land Algorithm. Missoula, MT: Univ. Montana, p. 57.

Jung, M, Schwalm, C, Migliavacca, M, Walther, S, Camps-Valls, G, Koirala, S, Anthoni, P, Besnard, S, Bodesheim, P, Carvalhais, N, Chevallier, F, Gans, F, Groll, D S, Haverd, V, Ichii, K, Jain, A K, Liu, J, Lombardozzi, D, Nabel, J E M S, . . . Walker, A. (2019). Scaling carbon fluxes from eddy covariance sites to globe: Synthesis and evaluation of the FLUXCOM approach. Biogeosciences Discussions, 2019, 1–40.

Running SW, Thornton PE, Nemani R, Glassy JM. 2000. Global Terrestrial Gross and Net Primary Productivity from the Earth Observing System. In: Sala O.E., Jackson R.B., Mooney H.A., Howarth R.W. (eds) Methods in Ecosystem Science. Springer, New York, NY.

Tramontana G, Jung M, Schwalm CR, Ichii K, Camps-Valls G, Ráduly B, Reichstein M, Arain MA, Cescatti A, Kiely G, Merbold L, Serrano-Ortiz P, Sickert S, Wolf S, Papale D. 2016. Predicting carbon dioxide and energy fluxes across global FLUXNET sites with regression algorithms, Biogeosciences, 13, 4291-4313.

Stevens NT, Steiner SH, MacKay RJ. 2017. Assessing agreement between two measurement systems: An alternative to the limits of agreement approach. Stat Methods Med Res. 26:2487-2504. (Jung et al., 2019)

---

## Referee Comment (RC2) · Anonymous Referee #2 · 28 Feb 2020

This paper looks at tower-based NPP estimates at a drylands site in Mexico and compares the results to MODIS NPP product. The authors find that the site is a net carbon sink and that the MODIS product underestimated GPP at this site. While these findings are interesting, the manuscript appears to lack a clear research question, and does not propose a way forward for this work: what are the large-scale implication for this site being a net carbon sink even though other similar sites are not? Can the MODIS product be combined with other data to improve the comparison with the in-situ data (beyond changes to the algorithm by the MODIS science team)? Has the MODIS product been used in other studies that are therefore obtaining biased results because they did not realize the issue with the MODIS product? Overall, the data is interesting but again, it seems like the analysis needs to be taken one step further before the manuscript is

publishable.

Introduction

There has been a lot of work recently on the importance of water-limited environments for carbon sequestration. It would be good to expand on current work, explain what the current hypothesis is for why savannas/drylands are thought to be so important for carbon sequestration and why this had been missed until recently. Properly embedding this work within this body of work would help raise its importance. After that, defining what the research question is, beyond "adding one more dataset" to the list, is missing.

Methods

Section 2.3 of the Methods gives the impression that the MODIS GPP product was consumed without a full understanding of how this product is generated from the MODIS data. Since the deviation of the MODIS product forms the in-situ data is at the core of the manuscript, it would be helpful to flesh out how GPP is estimated for the MODIS product. It would make the argument in the discussion stronger.

Results and discussion

Lumping results and discussion together often leads to a weaker discussion, and I think it is indeed the case here: splitting the two sections would allow the authors to expand on the broader impacts of the study, linking it back to other work, and further explaining the repercussions of MODIS' underestimation. The effect of the different photosynthesis mechanisms eluded to in the abstract would be interesting to further develop in the discussion as well. Finally, the link with carbon sink and overgrazing is alluded to but never actually discussed, even though it would be of interest to many other parts of the globe.

Specific comments:

L127: Why was Licor used only on a single occasion? Could the measurements have been repeated on a different day? L 138: which made difficult measuring their abundances

---

## Author Comment (AC2) · 23 Mar 2020

Dear reviewer,

Thank you for the comments and suggestions to improve the manuscript. Below we give some replies to the raised points, and we will carefully consider all of them in a revised manuscript. Your comments are included here along with our reply.

Reviewer: This paper looks at tower-based NPP estimates at a drylands site in Mexico and compares the results to MODIS NPP product. The authors find that the site is a net carbon sink and that the MODIS product underestimated GPP at this site. While these findings are interesting, the manuscript appears to lack a clear research question, and does not propose a way forward for this work: what are the large-scale implication

for this site being a net carbon sink even though other similar sites are not? Can the MODIS product be combined with other data to improve the comparison with the in-situ data (beyond changes to the algorithm by the MODIS science team)? Has the MODIS product been used in other studies that are therefore obtaining biased results because they did not realize the issue with the MODIS product? Overall, the data is interesting but again, it seems like the analysis needs to be taken one step further before the manuscript is publishable.

Response

There is ongoing work to combine different MODIS inputs along with onsite data to model gross primary production (GPP). One example is the ensemble of models from populations of solutions obtained using machine learning algorithms (Tramontana et al. 2016). We cited some of the literature where the agreement was not good between EC measurements and the MODIS algorithm. Some of these authors mention the problem with the MODIS land use classification.

The MODIS algorithm is based on the assumption that the radiation use efficiency of the vegetation, under well-watered and fertilized conditions, is linearly related to the amount of absorbed photosynthetically active radiation (APAR). A factor of radiation use conversion efficiency (epsilon) is used to multiply APAR and represent the actual productivity of the vegetation; epsilon varies by vegetation type and climate condition. This is the reason why the land use classification from MODIS is important, if vegetation is misclassified then the GPP estimate of MODIS would be biased.

A more detailed data analysis of EC data will be included. Site description and characterization using remote sensed data will be included from MODIS to contextualize the site within the study area. Estimates of GPP will be recalculated using the MODIS algorithm with adequate parameters for the vegetation type at Bernal site. We will use the available data layers from MODIS (Aqua and Terra) as inputs of a random forest regression ensemble for GPP to predict the GPP modeled from EC data (Tramontana

et al., 2016).

Reviewer: Introduction There has been a lot of work recently on the importance of water-limited environments for carbon sequestration. It would be good to expand on current work, explain what the current hypothesis is for why savannas/drylands are thought to be so important for carbon sequestration and why this had been missed until recently. Properly embedding this work within this body of work would help raise its importance. After that, defining what the research question is, beyond adding one more dataset to the list, is missing.

Response

Certainly, arid and semi-arid ecosystems have been considered as a source of carbon due to their low vegetation productivity. Among others, there are two hypotheses supporting carbon sequestration by these ecosystems: the role of the soil inorganic carbon and that of the CAM photosynthetic pathway. In either case, soil water is important for the carbon cycle, as are the intermittent rainfall pulses in their intensity and frequency. Small pulses would result in predominantly soil respiration while large pulses would reflect carbon absorption (Sun et al., 2017). For succulent CAM and C3 plants, stored water in their roots and photosynthetic stems confers the ability to grow and reproduce during intensely hot and dry periods; suggesting that carbon and water fluxes would be decoupled from soil water (Sandquist, 2014).

Soil is a key component in these ecosystems, because soil inorganic carbon (SIC) content can be considerably higher compared to the soil organic carbon (SOC) (Schlesinger, 1982). Caliche is calcium carbonate that has been leached out of bedrock or detrital material by meteoric water and precipitated by evaporation in the overlying soil zone; the caliche deposit occurs either as closely spaced laminae lying parallel to the ground surface, or as a cement that binds detrital fragments. The Bernal region is characterized by having caliche deposits with layers of 2-3 m thick (Segerstrom, 1961). Carbon dioxide may be taken up during the dissolution of soil

carbonates, but subsequent leaching of ionic calcium and bicarbonate to the lower soil profile, where carbonate precipitates, will result in a flux of carbon dioxide to the soil surface; when these deposits accumulate on non calcareous parent materials, they can represent a net sink for atmospheric carbon dioxide (Schlesinger, 2017). The interpretation of eddy covariance (EC) measurements in arid lands would benefit from measurements of changes in the SIC and then evaluate the contribution of non-biological processes and their potential as a carbon sink.

Much of the carbon uptake in water limited ecosystems is thought to be due to nocturnal carbon dioxide capture due to CAM metabolism (Osmond et al., 2008). Owen et al. (2016) estimated a 20.1 Mg dry biomass ha-1 y-1 yield for an Agave monoculture and identified a four phase alternating pattern of carbon dioxide sink at night and carbon dioxide source during the day. Cactus store massive amounts of water and other resources in the succulent roots, stems and leaves; these anatomic and physiological adaptations confer a remarkable ability to grow and reproduce during intensely hot and dry periods (Ogburn & Edwards, 2010). Cacti contain large quantities of calcium oxalate and when they decay these minerals are released and subsequently transformed to calcite and possibly later experience mobilization of colloidal complexes by movement of soil water (Garvie, 2006).

We will improve the manuscript with a deeper elaboration of the main question, which is the agreement between two methods of measurement of GPP: Eddy covariance and MODIS algorithm. To support this approach we propose: 1) Estimate GPP according to MODIS method but using the epsilon parameter and APAR calculated for our site (Heinsch et al., 2003; Running et al., 1999), 2) Using available data layers of MODIS (Aqua and Terra) and generate a random forest regression ensemble for eight-day GPP (Jung et al., 2019; Tramontana et al., 2019) and 3) Compare GPP values obtained by MODIS algorithm, MODIS with our data and modeled from the EC using a Bayesian approach (Stevens et al., 2018).

Reviewer: Methods Section 2.3 of the Methods gives the impression that the MODIS

GPP product was consumed without a full understanding of how this product is generated from the MODIS data. Since the deviation of the MODIS product forms the in-situ data is at the core of the manuscript, it would be helpful to flesh out how GPP is estimated for the MODIS product. It would make the argument in the discussion stronger.

Response

A full description of the MODIS algorithm will be included. Also there is the topic of ecosystem respiration that needs some clarification, regarding semiarid environments and the modeling of respiration from NEE data from EC and soil temperature, but not soil water.

Reviewer: Results and discussion Lumping results and discussion together often leads to a weaker discussion, and I think it is indeed the case here: splitting the two sections would allow the authors to expand on the broader impacts of the study, linking it back to other work, and further explaining the repercussions of MODIS underestimation. The effect of the different photosynthesis mechanisms eluded to in the abstract would be interesting to further develop in the discussion as well. Finally, the link with carbon sink and overgrazing is alluded to but never actually discussed, even though it would be of interest to many other parts of the globe.

Response

We will separate the results and discussion sections. In the discussion section we will delve further into the effect of overgrazing and the different photosynthetic pathways present in the examined vegetation on the carbon sink. While the herbaceous stratum was deemed as overgrazed, the shrubs and cactus had little utilization by cattle.

Specific comments: L127: Why was Licor used only on a single occasion? Could the measurements have been repeated on a different day?

Response

LAI measurements were used to describe the site conditions, not to support the hypothesis of the work, we could not make more measurements due to equipment availability. Instead we will include MODIS LAI, NDVI and EVI timeseries to complete the site characterization.

L 138: which made difficult measuring their abundances.

Response

The study site is a privately managed cattle ranch. Livestock grazed freely year round in the paddock. Grass species typically are identified by their inflorescence and leaf structures are difficult to differentiate between species when the vegetation is overgrazed, basically leaves are very small and grow close to the ground. An alternative would be to use wire-mesh exclosure cages to protect the plants from grazing and measure pasture accumulation rate along with the corresponding botanical dissection for plant identification (Radcliffe, 1982).

References

Garvie, L. A. (2006). Decay of cacti and carbon cycling. Naturwissenschaften, 93(3), 114–118.

Heinsch, F. A., Reeves, M., Votava, P., Kang, S., Milesi, C., Zhao, M., Glassy, J., Jolly, W. M., Loehman, R., & Bowker, C. F. (2003). Gpp and npp (mod17a2/a3) products nasa modis land algorithm. MOD17 User's Guide, 1–57.

Jung, M., Schwalm, C., Migliavacca, M., Walther, S., Camps-Valls, G., Koirala, S., Anthoni, P., Besnard, S., Bodesheim, P., Carvalhais, N., Chevallier, F., Gans, F., Groll, D. S., Haverd, V., Ichii, K., Jain, A. K., Liu, J., Lombardozzi, D., Nabel, J. E. M. S., . . . Walker, A. (2019). Scaling carbon fluxes from eddy covariance sites to globe: Synthesis and evaluation of the FLUXCOM approach. Biogeosciences Discussions, 2019, 1–40. https://doi.org/10.5194/bg-2019-368.

Ogburn, R. M., & Edwards, E. J. (2010). The ecological water-use strategies of succulent plants. En Advances in botanical research (Vol. 55, pp. 179–225). Elsevier.

Osmond, B., Neales, T., & Stange, G. (2008). Curiosity and context revisited: Crassulacean acid metabolism in the Anthropocene. Journal of experimental botany, 59(7), 1489–1502.

Radcliffe, J. (1982). Effects of aspect and topography on pasture production in hill country. New Zealand Journal of Agricultural Research, 25(4), 485–496.

Running, S. W., Nemani, R., Glassy, J. M., & Thornton, P. E. (1999). MODIS daily photosynthesis (PSN) and annual net primary production (NPP) product (MOD17) Algorithm Theoretical Basis Document. https://www.researchgate.net/publication/237110406.

Sandquist, D. (2014). Plants in deserts. Ecology and the Environment. The Plant Sciences.(8), 297–326.

Schlesinger, W. H. (1982). Carbon storage in the caliche of arid soils: A case study from Arizona. Soil Science, 133(4), 247–255.

Schlesinger, W. H. (2017). An evaluation of abiotic carbon sinks in deserts. Global change biology, 23(1), 25–27.

Stevens, N. T., Rigdon, S. E., & Anderson‐Cook, C. M. (2018). Bayesian probability of predictive agreement for comparing the outcome of two separate regressions. Quality and Reliability Engineering International, 34(6), 968–978.

Sun, Q., Meyer, W. S., Koerber, G. R., & Marschner, P. (2017). Prior rainfall pattern determines response of net ecosystem carbon exchange to a large rainfall event in a semi-arid woodland. Agriculture, Ecosystems & Environment, 247, 112–119.

Tramontana, G., Jung, M., Keenan, T., Migliavacca, M., Reichstein, M., Ogee, J., Camps-Valls, G., & Papale, D. (2019). A machine learning based approach for estimating gross carbon dioxide fluxes from eddy covariance net ecosystem exchange

measurements. 21.

---

## Author Response (AR1)

**Final response and modified manuscript**

Dear Editor,

Please consider the revised manuscript incorporating the reviewers suggestions. Our responses follow each of the main reviewers comments which are inserted in boldface:

**Reviewer 1**

**The manuscript appears to lack a clear research question**

A modified introduction section presents a more complete background to frame the research questions related to the identified problems.

**Can the MODIS product be combined with other data to improve the comparison with the in-situ data (beyond changes to the algorithm by the MODIS science team)?**

We present alternative models to represent gross primary production (GPP) with environmental and MODIS data.

**Has the MODIS product been used in other studies that are therefore obtaining biased results because they did not realize the issue with the MODIS product?**

The discussion section makes a reference to studies in semi-arid ecosystems and highlights those papers identifying the probable causes of disagreement with the MODIS GPP estimates.

**Overall, the data is interesting but again, it seems like the analysis needs to be taken one step further before the manuscript is publishable.**

The revised work uses ensembles of machine learning, algorithms using automated procedures, as well as a recently published procedure to compare two measurement systems with emphasis on accounting for both bias and variability. The results section shows how to adjust MODIS estimates using the parameters of the probability of agreement function.

**It would be good to expand on current work, explain what the current hypothesis is for why savannas/drylands are thought to be so important for carbon sequestration and why this had been missed until recently.**

The introduction was re-worked to present the role of semi-arid environments in the carbon cycle and the relation to management practices.

**Methods Section 2.3 of the Methods gives the impression that the MODIS GPP product was consumed without a full understanding of how this product is generated from the MODIS data. Since the deviation of the MODIS product forms the in-situ data is at the core of the manuscript, it would be helpful to flesh out how GPP is estimated for the MODIS product. It would make the argument in the discussion stronger.**

Now there is a section describing the MODIS GPP algorithm and the important assumptions that need to be revised.

**Results and discussion** Lumping results and discussion together often leads to a weaker discussion, and I think it is indeed the case here: splitting the two sections would allow the authors to expand on the broader impacts of the study, linking it back to other work, and further explaining the repercussions of MODIS underestimation. The effect of the different photosynthesis mechanisms eluded to in the abstract would be interesting to further develop in the discussion as well. Finally, the link with carbon sink and overgrazing is alluded to but never actually discussed, even though it would be of interest to many other parts of the globe.

Results and discussion are in different sections. The topics suggested were included.

**Why was Licor used only on a single occasion? Could the measurements have been repeated on a different day?**

Information using the Li-6400 was eliminated since sampling was incomplete and the overall contribution to the paper was poor.

**Reviewer 2**

**Introduction: The introduction lacks a clear scientific question and related hypothesis. Providing new measurements of NEE at underrepresented ecosystems is important, as well as comparing GPP estimates with satellite-derived products. That said, this manuscript should emphasize what is new (beyond new measurements) and have a Discussion paper testable hypothesis if possible. The introductions resemble a technical report and could be improved by framing it around a clear scientific question.**

Background information required to better understand the paper was included and the hypothesis was explicitly presented: The Bernal site was a carbon sink? What was the agreement between remote sensed GPP from MODIS and measurements from the EC tower? Alternative methods to estimate GPP were included.

**Methods: This section requires substantial reorganization and more information. The authors should link the methods to the research questions/hypotheses, provide more information about the site and how data was processed and analyzed. Maybe a section about data analysis would help to improve this section.**

The Methods section was reorganized and different subsections were included. A more complete description of the different analysis was presented.

**Results/Discussion section: I strongly recommend separating results from the discussion. Without a clear scientific question and testable hypothesis, it is difficult to evaluate this section and the novelty of the results. The authors touch different topics from leaf level photosynthesis, ecosystem level fluxes and remote sensing but I feel that there is disconnection between the results in this section. Finally, due to the limited dataset and analyzes, this section seems to be over-interpreting the results and consequently I wonder if this manuscript is premature for this study site.**

Two separate sections were presented. Data that was not relevant to the study was removed. In a previous response we explained the reasons why we can't present a more complete dataset, mainly the site suffered the clearing of all vegetation during 2019.

**Conclusion: I believe that this section is not fully supported by the data analysis and results. Again, it is difficult to evaluate this manuscript as the authors touch several topics, but none is analyzed in detail leaving the manuscript presenting a very broad (and potentially over interpreting) view of results. I respectfully believe that this is a good first step to summarize results from this study site, but this study requires substantial improvements in quality and quantity of data (e.g., longer datasets), conceptual organization (e.g., questions/hypotheses), and further analyzes to test clear hypotheses to provide a novel scientific contribution.**

**Eddy covariance carbon flux in a scrub in the Mexican highland**

Aurelio Guevara-Escobar[1], Enrique González-Sosa[2], Mónica Cervantes-Jiménez[1], Humberto Suzán-Azpiri[1], Mónica Elisa Queijeiro-Bolaños[1], Israel Carrillo-Ángeles[1], Víctor Hugo Cambrón-Sandoval[1].

[1]Facultad de Ciencias Naturales, Universidad Autónoma de Querétaro, Av. de las Ciencias s/n Juriquilla CP. 76230, Querétaro, Querétaro.
[2]Facultad de Ingeniería, Universidad Autónoma de Querétaro. Cerro de las Campanas s/n Las Campanas, CP. 76010 Querétaro, Querétaro.

*Correspondence to*: Mónica Cervantes-Jiménez (monica.cervantes@uaq.mx)

**Abstract.** Arid and sem-iarid ecosystems contain relatively high species diversity and are subject to intense use, in particular extensive cattle grazing which has favoured the expansion and encroachment of perennial thorny shrubs into the grasslands, thus decreasing the value of the rangeland. However, these environments have been shown to positively impact global carbon dynamics. The Moderate Resolution Imaging Spectroradiometer (MODIS) gross primary productivity (GPP) product provides a rapid and broad-scale means for monitoring rangelands, but few studies have validated the performance of MODIS estimates in arid and semi-arid ecosystems. We measured the net ecosystem exchange of C (NEE) with the Eddy Covariance (EC) method and estimated GPP in a thorny scrub at Bernal in Mexico. The hypothesis was that this site might behave as carbon sink. We also tested the agreement with remote sensed GPP estimates from MODIS model. The agreement with EC estimates of two alternative modeling methods were tested: ordinary least squares multiple regression (OLS) or ensembles of machine learning algorithms (EML); the variables used as predictors were MODIS spectral bands, vegetation indices and environmental variables. The Bernal site was a carbon sink despite it is being in an overgrazed condition, the average NEE during fifteen months of 2017 and 2018 was -0.78 g C m$^{-2}$ d$^{-1}$ and the flux was negative or neutral during the measured months. The probability of agreement was higher for the EML (0.6) followed the OLS (0.5) and then MODIS (0.24). We used an EML from a site with similar vegetation and climate to predict GPP at Bernal but the probability of agreement was poor (0.16), indicating the local specificity of this model. Although cacti were an important component of the vegetation the nighttime flux was characterized by positive NEE suggesting that the photosynthetic dark-cycle flux of cacti was lower than ecosystem respiration. The discrepancy between MODIS and EC GPP estimates stresses the need to understand the limitations of both methods.

**Comentado [R1]:** The abstract has been modified to include new results and changes in the paper.

[revised manuscript text omitted]

**Comentado [R2]:** This part is the introduction to attend the question of the reviewer 1: **"Can the MODIS product be combined with other data to improve the comparison with the in-situ data (beyond changes to the algorithm by the MODIS science team)?"**

To generate models of GPP we measured EC fluxes during 2017-2018 in a thorny scrub with semi-arid climate in the highlands of Mexico (Bernal site). Competing models were data-driven machine learning regression ensembles (EML) and ordinary least squares regression (OLS), both using Daymet (Thornton et al., 2017) and MODIS data sets as explanatory variables. The MODIS GPP product was used as a baseline comparison. The second step was to use a EML model based on

5 local data (Daymet and MODIS) from a site with EC instrumentation and similar vegetation to that of Bernal's site and then use that model to predict GPP at the Bernal site. The site we used was Santa Rita from the Ameriflux network. While Santa Rita is in the Sonoran Desert and Bernal is in the southern border of the Chihuahuan desert, both have a similar climate and mesquite vegetation (Figure 1). A good agreement between Bernal EC data and the predictions from Santa Rita model, would support the use of machine learning algorithms as a scale up mechanism. This would be useful to the understanding of

10 rangelands and also improve current earth system models (Piao et al., 2019).

We hypothesized whether or not the semi-arid vegetation at Bernal site was a carbon sink during the wet season or a carbon source during the dry season, since some of the site species reproduce during winter and spring, particularly cacti, *Acacia* and *Prosopis* (Mesquite). Furthermore, the Bernal site had a history of disturbance by overgrazing, this could decrease the GPP and even result in a positive carbon balance; thus being a carbon source. On the contrary, confirming the hypothesis

15 that the shrub vegetation in this semi-arid environment is a carbon sink would contribute to reinforce the reported importance of semi-arid environments in the global carbon balance (Zhang et al., 2020). The measurement campaign presented here was short (March 30 2017 to August 22 2018) but it was not biased by wet winters, since both years were characterized by a less than weak Niño-Niña.

**2 Materials & Methods**

20 ### 2.1 Site description

The study site (Bernal) is located at N 20,717, W 99,941 and 2 050 m a.s.l. in the municipality of Ezequiel Montes in Querétaro where real estate development, feedlot beef production, cheese and wine production associated with tourism and, automotive industry development are very attractive options for landowners in the region. Bernal is located in a shallow valley oriented from north to south, approximately 15 to 20 km wide and opening to the south to the Río Lerma basin and

25 then draining into the Pacific Ocean. The northern limit of the valley is surrounded by hill country and its characteristic 433 m in height dacitic dome (Aguirre-Díaz et al., 2013). Moisture-laden winds blow westward from the Gulf of Mexico but the Sierra Gorda, located 60 km east of Bernal, casts a rain shadow over the area (Segerstrom, 1961).

**Comentado [R3]:** In this part we made an introduction to clarify the context and hypotesis:
**Overall, the data is interesting but again, it seems like the analysis needs to be taken one step further before the manuscript is publishable.**

**Comentado [R4]:** We modified the introduction and include an explicit hypothesis to address reviewer 1's comment:
**It would be good to expand on current work, explain what the current hypothesis is for why savannas/drylands are thought to be so important for carbon sequestration and why this had been missed until recently.**

**Comentado [R5]:** We improve this section to attend the comment of the reviewer 2:
**Introduction: The introduction lacks a clear scientific question and related hypothesis. Providing new measurements of NEE at underrepresented ecosystems is important, as well as comparing GPP estimates with satellite-derived products. That said, this manuscript should emphasize what is new (beyond new measurements) and have a Discussion paper testable hypothesis if possible. The introductions resemble a technical report and could be improved by framing it around a clear scientific question.**

**Comentado [R6]:** The Methods section was reorganized and different subsections were included. A more complete description of the different analysis was presented attending the comment of the reviewer 2:
**Methods: This section requires substantial reorganization and more information. The authors should link the methods to the research questions/hypotheses, provide more information about the site and how data was processed and analyzed. Maybe a section about data analysis would help to improve this section.**

[revised manuscript text omitted]

**Comentado [R8]:** This section was divided in two, but did not undergo major changes, only the wording changed a little.

**2.4 Remote sensed data**

Data was requested via the land processes DAAC AppEEARS to obtain spatial and temporal subsets for the Bernal and Santa Rita sites including: daily surface reflectance (MOD09GA.006 and MODOCGA.006), daily day and night time land surface temperature (LST) (MOD11A1.006 and MYD11A1.006), eight day leaf area index (LAI) and fraction of photosynthetically active radiation (FPAR) (MOD15A2H.006, MCD15A2H.006), sixteen day enhanced vegetation index (EVI) (MOD13Q1.006, MYD13Q1.006), sixteen day gross primary production (GPP) and net photosynthesis (PsnNet) (MOD17A2H.006). Appendix A presents the details of each spectral band of MODIS. Data with less than good quality flags were deleted. Missing data was filled with splines and a database with one day time step was generated, this would smooth linear temporal phenological evolution between any two successive remotely sensed data points (Eshel et al., 2019). Daily accumulated rainfall was requested using the Giovanni GSFC platform (GPM 3IMERGDF.006 and TRMM 3B42.007). Gridded weather parameters from de ORNL DAAC Daymet dataset were: precipitation, shortwave radiation, maximum and minimum air temperature and water vapor pressure. Daymet is a data product derived from a collection of algorithms interpolating and extrapolating daily meteorological observations (Thornton et al., 2017). Following Henrich et al. (2012) and Hill et al. (2006), daily reflectance bands of MODIS were used to compute several vegetation indices: Red Green Ratio Index (RGRI), Simple Ratio (SimpleR), Moisture Stress (MoistS), Disease Stress Index (DSI), Normalized Difference Vegetation Index (NDVI), Normalized Difference Water Index (NDVI_w) and Enhanced Vegetation Index (EVI); the corresponding equations are presented in Appendix A.

> **Comentado [R9]:** This section was added to describe the MODIS remote sensor data used in modelling.

[revised manuscript text omitted]

**Comentado [R10]:** These sections were added to attend the reviewer's comments:

**Reviewer1:** Methods Section 2.3 of the Methods gives the impression that the MODIS GPP product was consumed without a full understanding of how this product is generated from the MODIS data. Since the deviation of the MODIS product forms the in-situ data is at the core of the manuscript, it would be helpful to flesh out how GPP is estimated for the MODIS product. It would make the argument in the discussion stronger.

**Reviewer 2:** Methods: This section requires substantial reorganization and more information. The authors should link the methods to the research questions/hypotheses, provide more information about the site and how data was processed and analyzed. Maybe a section about data analysis would help to improve this section.

**Comentado [R11]:** Results and discussion was separated accord to reviewer's 1 and 2 suggestions:

**Reviewer1:** Results and discussion Lumping results and discussion together often leads to a weaker discussion, and I think it is indeed the case here: splitting the two sections would allow the authors to expand on the broader impacts of the study, linking it back to other work, and further explaining the repercussions of MODIS underestimation. The effect of the different photosynthesis mechanisms eluded to in the abstract would be interesting to further develop in the discussion as well. Finally, the link with carbon sink and overgrazing is alluded to but never actually discussed, even though it would be of interest to many other parts of the globe.

**Reviewer 2:** Results/Discussion section: I strongly recommend separating results from the discussion. Without a clear scientific question and testable hypothesis, it is difficult to evaluate this section and the novelty of the results. The authors touch different topics from leaf level photosynthesis, ecosystem level fluxes and remote sensing but I feel that there is disconnection between the results in this section. Finally, due to the limited dataset and analyzes, this section seems to be over-interpreting the results and consequently I wonder if this manuscript is premature for this study site.

**Comentado [R12]:** This paragraph was added to introduce the results.

[revised manuscript text omitted]

**Comentado [R16]:** This section was added to show the agreement between GPP modelled with Ensemble of Machine Learning (EML), Ordinary Least Squares (OLS) and MODIS.

**Comentado [R17]:** This section was modified attending to the comment of the reviewer 1: "Has the MODIS product been used in other studies that are therefore obtaining biased results because they did not realize the issue with the MODIS product?"

[revised manuscript text omitted]

---

## Author Response (AR2)

**Final response and modified manuscript**

Dear Editor,

Please consider the revised manuscript incorporating the reviewers suggestions. Our responses follow each of the main reviewers comments, which are inserted in boldface:

**Reviewer #2**

**Overall, this new version of the paper is much stronger than the last one. The authors have considerably expanded their analysis, now including a comparison of various regression models to predict GPP. The Bernal site now becomes a study site for a study with a much broader impact for the community. The comparison between EC and MODIS is very interesting and makes a compelling case for a more careful use of remotely sensed GPP in semi-arid grasslands.**

**I do think that the paper could be strengthened by clearly embracing the GPP comparison. Indeed, I find that the title, abstract, and even to some degree the introduction are not fitting this new version of the paper anymore, and might actually do a disfavor to the article, making these new interesting results more difficult to find. I think that rewriting the abstract and introduction to focus on the GPP comparison and thinking of the Bernal site as a study site would, in the end, be a more powerful narrative here, and be a better fit to the results and discussion sections as they are.**

We restructured the work and improved the wording to emphasize the comparison of machine learning and automated procedures methods for predicting GPP, as well as a recently published procedure to compare two measurement systems with emphasis on accounting for both bias and agreement. We modified the introduction and the abstract to focus on the findings of our work.

**I have a few other minor comments below:**

**Page 1, L16: remotely sensed**

It has been replaced.

**L19 despite it being in an overgrazed condition**

It has been replaced.

**Page 2, L3: rural-urban migration being an important driver…**

It has been replaced.

**Page 3, L13: what's is "inter-converted"?**

The phrase was restructured.

**Page 5 L33: do we know why the shrub was cleared?**

Vegetation was removed to increase grazing area for cattle.

**Page 5, L35: two verbs in one sentence "The soil has a clay loam texture are Vertisols with abundant sub rounded basaltic stones…", please rephrase**

The phrase was restructured.

**Page 6, L2: the vegetation corresponds to**

Thank you. It has been replaced.

**P17, L19: for the Bernal site**

Thank you. It has been replaced.

**P17, L21: gurant? Typo**

Thank you. It has been replaced.

**P17, L21-23: However, during… exchange at that time: This sentence is grammatically incorrect and confusing, please rephrase.**

Thank you. It has been replaced.

**P22, L4: predicted GPP**

It has been replaced.

**P23, L31: complicates…the calculation?**

It has been replaced.

**Section 4.3 (pages 24-25) seems like it really belongs to the Results section, and not the discussion**

This section was reorganized to better focus on the discussion.

**Section 4.4 looks to me like the "discussion" part of "result" section 4.3**

The results and discussion sections were restructured to emphasize the findings obtained and topics out of context were omitted. The names of the sections were changed to improve the reading sequence.

**Page 25, L25-26: rephrase this sentence, it's difficult to understand**

The phrase was restructured.

**Referee #3**

**Guevara-Escobar measure eddy covariance fluxes from a scrub ecosystem in Mexico and explore the ability of machine-learning and other models, including MODIS, to predict measured GPP. Many aspects of the analysis are quite rigorous, but the manuscript needs comprehensive improvement to explain why particular analyses were used, why they are important (the Introduction needs to be comprehensively rewritten) and how to interpret results (the Discussion often reads like a qualitative Results section rather than a synthesis of what was learned). Other parts of the manuscript like the abstract and for the most part the Methods were informative and well-written. By focusing on the major ideas and clearly communicating the important parts of the analysis, the manuscript will make an interesting contribution to the literature. I also have some concerns about how GPP was estimated given that the u\* threshold seems anomalously low. More details on this would be forthcoming.**

Thanks for your annotations and comments. We followed your comments to y restructure the presentation, particularly in the introduction, results, discussion and conclusion as to highlight the purpose of the work and the findings found.

**The statement on page 1 line 28 could use a reference, I have seen other publications that have put this value at 40%. It could also be combined with the next statement to read '...ecosystems, but almost 2 million km2 (50%) in Mexico, mainly the...'**

Thanks, the phrase was restructured and the reference of Verbist et al., 2010 was added to support this estimate.

**The first paragraph of the as a whole should be rewritten to be more simple and lead more clearly from the geographic extent of semi-arid and arid ecosystems to their degradation. Breaking the paragraph into 2 paragraphs will help.**

The paragraph was restructured and split in two parts as you suggested.

**Delete this statement, it is unimportant in this context: Substantial biosphere–precipitation feedback is often found in regions that are transitional between energy and water limitation, such as semi-arid or monsoonal regions (Green et al., 2017). The introduction as a whole sounds like a literature review in need of more structuring. Move from general information to the specific problem of measuring GPP in degraded semiarid ecosystems. At the moment there is too much information and it is confusing to read.**

The introduction section was restructured to focus on the importance of semi-arid ecosystems and their estimation of GPP.

**For the site description, I assume that 20,717 means 20.717, i.e. decimal degrees?**

Yes, you are right, we replace de comma by a point.

**'We assumed that the footprint of the EC was measuring only the patchy scrub vegetation'. This can be tested and non-target-ecosystem measurements removed. Alternately, it may not be important if structures are sufficiently far away. How far away is far away?**

Thanks for the annotation. We added the figure 2 with the Eddy Covariance footprint of the Bernal site.

**Was the u\* threshold really 0.033 m/s? This is very low, probably too low, but I'm also confused about the wording that describes this application. I am concerned that many fluxes with insufficient turbulence are being included in the final output, which would cause respiration to be too low and carbon uptake estimates to be too high. GPP estimates would then be too high which may be why the fit in Figure 6 is so poor. How do the GPP estimates compare against other ecosystems? This is a topic for the Results section and I am mostly interested in knowing if the GPP results are defensible rather than a comprehensive comparison of GPP at semi-arid ecosystems.**

The wording was changed and revised for clarity, also, Appendix A was added with the u\* values above which  fluxes were considered as valid. In the previous version the sentence "The filled-in estimates of NEE (NEE_uStar_f), GPP (GPP_uStar_f), and $R_{eco}$ (Reco_uStar) were used from the u\* annual base scenario because the difference between the base u\* scenario and the 95% u\* uncertainty threshold was 0.033 m s$^{-1}$; only 8.5% of half hour records had a u\* below the 95% u\* threshold." was not meant to specify a threshold, but to explain that the threshold used was the base u\* scenario.

**eq. 1 and 2 (and 3): please use the multiplication sign and not the star in formal equations.**

We changed the multiplication sign in equations.

**I happen to find the model comparison with Santa Rita interesting: how much variability at a site can be explained by a model from another site?**

This comment is attended in the last paragraph of the 3.5 Agreement section.

**page 10 line 10: the authors of the R package should be cited.**

Thank you, but H2O.ai is a corporate author of the  open source R package,  the reference is correct.

**The first paragraph of the Results should be deleted. Perhaps as a first paragraph of a Discussion section but even then probably unneccesary.**

Thank you, the results and discussion sections were restructured to emphasize the findings obtained and topics out of context were omitted. The names of the sections were changed to improve the reading sequence. And the initial paragraph in each section were deleted as you suggested.

**Section 3.4 should come earlier in the results section. Reviewers need to be convinced that fluxes are defensible before modeling is described.**

We move the flux carbon results section at the first part of results.

**The IVI analysis (Table 3) would be more convincing if it was placed in the context of the flux footprint. Do these numbers come from across the entire site including wind directions that are uncommon?**

The figure 2 shows the tower footprint and the points of vegetation sample plots. This points were distributed across the entire site.

**Section 4.1: how would one interpret these results and why was EML best?**

In the 3.5 Agreement section of results, we added the confidence intervals to show the differences between the agreement of the methods used to modeling GPP.

**'is generally considered to be 1 km2' is incorrect. The footprint dimensions depend on sensor height and the Obukhov length.**

This sentence was deleted.

**Section 4.3 is interesting but discussing water leakiness without measurements is unnecessary.**

Thank you, the results and discussion sections were restructured to emphasize the findings obtained and topics out of context were omitted.

**4.4 call this section 'overgrazing' or 'ecosystem management' instead. Discussing erosion is outside of the context of the manuscript. The manuscript has too much information, much of it extraneous.**

The section was renamed "Ecosystem management".

**5. Conclusions: reasons why machine learning fit best and what can be learned from this for ecosystem analysis are still lacking.**

Thank you, the conclusion was rephrased to emphasize the findings obtained and what was learned along the work process.

Best Regards

Mónica Cervantes-Jiménez

[revised manuscript text omitted]

**Comentado [R7]:** Section 3.4 should come earlier in the results section. Reviewers need to be convinced that fluxes are defensible before modeling is described.

**Comentado [R8R7]:** Thanks for the annotation. We move the carbon flux section to the first part of results.

[Figure]

[revised manuscript text omitted]

[revised manuscript text omitted]

Comentado [R11]: Rv2: I happen to find the model comparison with Santa Rita interesting: how much variability at a site can be explained by a model from another site?

Comentado [R15]: Section 4.1: how would one interpret these results and why was EML best?

Comentado [R16R15]: Section 3.5 in Results, was added to attend this comment.

[revised manuscript text omitted]

---

## Author Response (AR3)

**Final response**

Dear Editor,

Please consider the revised manuscript incorporating the reviewer suggestions and comments.

**Reviewer comment:**

**The manuscript as written represents an improvement and I note that this review refers to the technical aspects of the analysis; copyediting is necessary and various aspects of the flow of the manuscript could still be improved. But the comparison between the machine learning and "standard" methods for carbon flux estimation is rigorous and interesting and I recommend that the manuscript be accepted following technical (mostly language) corrections for this reason.**

**Figure 2: is this the 90% flux footprint or the peak of the source weight function or something else? I'm not sure what the gray dots refer to. The conclusion can be strengthened by noting the Santa Rita site in comparison, which further emphasizes the challenge that ML approaches may be best applied locally and that future research should seek ways to make their predictions more generalizable (ideally through a mechanistic understanding of ecosystem processes).**

Many thanks for your reviews and dedication. Please consider this new version. We attended the main two comments regarding to Figure 2 and conclusion. Other minor corrections were made as you suggested.

If you have any other comments, we will happy to solve them.

Best Regards

Mónica Cervantes-Jiménez